

# 1 Understanding the Optical Properties of Ambient

# 2 Sub- and Supermicron Particulate Matter: Results from

# 3 the CARES 2010 Field Study in Northern California

Christopher D. Cappa[1,*], Katheryn R. Kolesar[1,+], Xiaolu Zhang[1], Dean B. Atkinson[2], Mikhail S. Pekour[3],
Rahul A. Zaveri[3], Alla Zelenyuk-Imre[3], Qi Zhang[4]
[1] Department of Civil and Environmental Engineering, University of California, Davis, CA 95616, USA
[2] Department of Chemistry, Portland State University, Portland, OR, 92707, USA
[3] Atmospheric Sciences & Global Change Division, Pacific Northwest National Laboratory, Richland,
WA, 99352, USA
[4] Department of Environmental Toxicology, University of California, Davis, CA 95616, USA
[+] Now at Department of Chemistry, University of Michigan, Ann Arbor, MI 48109, USA
*Correspondence to: Christopher D. Cappa (cdcappa@ucdavis.edu)

## 13 Short Summary

One way in which particles impact the solar radiation budget is through absorption and scattering
of solar radiation. Here, we report on measurements of aerosol optical properties at visible wavelengths
in the Sacramento, CA region and characterize their relationships with and dependence upon particle
composition, particle size, photochemical ageing, water uptake and heating.

## 18 Abstract

Measurements of the optical properties (absorption, scattering and extinction) of $PM_1$, $PM_{2.5}$ and
$PM_{10}$ made at two sites around Sacramento, CA during the June 2010 Carbonaceous Aerosols and
Radiative Effects Study (CARES) are reported. These observations are used to establish relationships



between various intensive optical properties and to derive information about the dependence of the optical
properties on photochemical ageing and sources. Supermicron particles contributed substantially to the
total light scattering at both sites, about 50% on average. A strong, linear relationship is observed between
the scattering Ångstrom exponent for $PM_{10}$ and the fraction of the scattering that is contributed by
submicron particles ($f_{sca,PM1}$) at both sites and with similar slopes and intercepts (for a given pair of
wavelengths), suggesting that the derived relationship may be generally applicable for understanding
variations in particle size distributions from remote sensing measurements. At the more urban T0 site, the
$f_{sca,PM1}$ increased with photochemical age whereas at the downwind, more rural T1 site the $f_{sca,PM1}$
decreased slightly with photochemical age. This difference in behavior reflects differences in transport,
local production and local emission of supermicron particles between the sites. Light absorption is
dominated by submicron particles, but the there is some absorption by supermicron particles (~15% of
the total). The supermicron absorption derives from a combination of black carbon that has penetrated
into the supermicron mode and from dust, and there is a clear increase in the mass absorption coefficient
of just the supermicron particles with increasing average particle size. The mass scattering coefficient
(MSC) for the supermicron particles was directly observed to vary inversely with the average particle
size, demonstrating that MSC cannot always be treated as a constant in estimating mass concentrations
from scattering measurements, or vice versa. The total particle backscatter fraction exhibited some
dependence upon the relative abundance of sub- versus supermicron particles, however this was
modulated by variations in the median size of particles within a given size range; variations in the
submicron size distribution had a particularly large influence on the observed backscatter efficiency and
an approximate method to account for this variability is introduced. The relationship between the
absorption and scattering Ångstrom exponents is examined and used to update a previously suggested
particle classification scheme. Differences in composition led to differences in the sensitivity of $PM_{2.5}$ to
heating in a thermodenuder to the average particle size, with more extensive evaporation (observed as a
larger decrease in the $PM_{2.5}$ extinction coefficient) corresponding to smaller particles, i.e. submicron
particles were generally more susceptible to heating than the supermicron particles. The influence of
heating on the particle hygroscopicity varied with the effective particle size, with larger changes observed
when the $PM_{2.5}$ distribution was dominated by smaller particles.



## 1. Introduction


Atmospheric aerosol particles impact regional and global climate by scattering and absorbing solar
radiation, as well as by affecting the properties of clouds, although the magnitude of these impacts remain
uncertain (IPCC, 2013). The specific ability of particles to interact with solar radiation is dependent upon
particle size, morphology and composition (Bohren and Huffman, 1983), which are often linked through
variations in sources and chemical processing. Atmospheric particles span a wide range of sizes, from just
a few nm to many microns. Quantitative understanding of the absolute and relative contributions across
this entire size range is necessary to assess their climate impacts (Schwartz, 1996). The major sources of
particles within the submicron and supermicron size regimes differ, with submicron particles generally
deriving from combustion emissions and secondary formation and supermicron particles generally
coming from mechanical action (e.g. wind-blown dust or ocean wave breaking) (Seinfeld and Pandis,
1998). As such, particle composition varies across this size range, as does the effectiveness with which
particles absorb and scatter solar radiation. *In situ* measurements of the wavelength- and size-dependent
light scattering and absorption properties of ambient atmospheric particles made concurrent with
measurements of the size-dependent particle composition can provide insights into the impacts of particles
on local climate and air quality (Anderson et al., 2003; Jung et al., 2009), as well as into the more general
relationships between particle size, composition and radiative effects that determine their global impacts
(Quinn et al., 2004; Bates et al., 2006; Wang et al., 2007; Garland et al., 2008; Gyawali et al., 2009; Yang
et al., 2009). Such *in situ* measurements can help to interpret observations from remote sensing (Russell
et al., 2010; Giles et al., 2012) and to provide observational constraints for results from simulations using
regional and global models (Kaufman et al., 2002; Myhre et al., 2012; Tsigaridis et al., 2014).
The US Department of Energy Carbonaceous Aerosols and Radiative Effects Study (CARES) took
place in June 2010 with a motivation of improving our understanding of aerosol optical properties and
how they evolve in the atmosphere through observations (Zaveri et al., 2012). Two heavily-instrumented
ground observational sites were set up, one within the greater Sacramento, CA urban area and one
approximately 40 km to the northeast in much more rural Cool, CA (Figure S1). At both sites a variety of
aerosol particle physical, chemical and optical property measurements were made. These two sites were
chosen because of the generally reproducible wind patterns that, much of the time, bring air up from the
San Francisco Bay Area (~100 mi SW) and past the Sacramento urban core before continuing up towards
the foothills of the Sierra Nevada mountains where the air mass accumulates biogenic emissions during
the day, with a reversed flow at night bringing more biogenically influenced air down from the mountains.
In this way, comparison between the two sites facilitates understanding of the role that atmospheric





photochemical processing plays in altering particle optical properties. Results from measurements of dry
particle light scattering and absorption made for submicron particulate matter ($PM_1$) and for PM smaller
than 10 μm ($PM_{10}$) at both sites and of dry particle light absorption and extinction for PM smaller than
2.5 μm ($PM_{2.5}$) at just the urban Sacramento site are reported in this study. The separate $PM_1$ and $PM_{10}$
measurements allows for determination of the optical properties of both submicron and supermicron (PM
> 1 μm) particles. The distinct sub- and supermicron measurements are used here to characterize and
examine the variability in their relative contributions as well as the differences between their properties
and sources in the summertime Sacramento region. These regional results are also used to develop more
general understanding of the size- and composition-dependent variability in aerosol particle optical
properties. The analysis here focuses especially on quantifying and assessing the relationships between
various intensive optical parameters (such as the scattering Ångstrom exponent, the absorption Ångstrom
exponent, and the single scatter albedo) or between these parameters and characteristics of the particle
distribution (such as the fine mode fraction or characteristic particle diameter) and how these differ
between size ranges (sub- versus supermicron) or are influenced by photochemical ageing or heating.

## 2. Experimental

All measurements were made during the 2010 CARES study, which took place in the Sacramento,
CA region from 2-29 June, 2010. Measurements were made at one of two sites: one located in the greater
Sacramento urban region (termed the T0 site) and one located ca. 40 km northeast downwind in Cool, CA
(termed the T1 site), shown in Figure S1. The CARES study has previously been described in detail (Fast
et al., 2012; Zaveri et al., 2012), and only a brief overview is given here. A list of instrumentation used in
this study is given in Table 1.

### 2.1 Measurements at the T0 Site

Particle light absorption coefficients ($b_{abs}$) were measured using the UC Davis (UCD) photoacoustic
spectrometer (PAS) at 405 nm and 532 nm (Lack et al., 2012) and using a particle soot absorption
photometer (PSAP; Radiance Research, Inc.) at 470 nm, 532 nm and 660 nm (Bond et al., 1999; Virkkula
et al., 2005). The PAS measured light absorption for $PM_{2.5}$ sampled through a cyclone, alternating on a
2.5 or 5 minute time scale between bypass measurements (i.e. dried ambient particles) and measurements
behind a constant-temperature (225°C) thermodenuder (Huffman et al., 2008; Cappa et al., 2012) with a
residence time of 5 s. The PAS was calibrated before, during and after the study by referencing the
observed photoacoustic response to added ozone to the extinction measured at the same wavelengths via
cavity ringdown spectroscopy (CRDS) (Lack et al., 2012). The PSAP measured $b_{abs}$ for $PM_1$ or $PM_{10}$ on





an alternating 6 minute cycle. The PSAP was corrected for spot size, flow and particle scattering using
standard methods (Bond et al., 1999; Ogren, 2010). Light scattering and backscattering coefficients ($b_{sca}$
and $b_{bsca}$, respectively) were measured for dried particles using a three-wavelength total/backscatter
nephelometer (TSI, Model 3563) at 450 nm, 550 nm and 700 nm. The nephelometer sampled $PM_1$ and
$PM_{10}$ on the same alternating 6 minute cycle as the PSAP. The $b_{sca}$ and $b_{bsca}$ were corrected for truncation
error (Anderson and Ogren, 1998). Light extinction coefficients ($b_{ext} = b_{abs} + b_{sca}$) were directly measured
for $PM_{2.5}$ using the UCD aerosol CRDS (Langridge et al., 2011; Cappa et al., 2012). The $b_{ext}$ were
measured for dried particles at both 405 nm and 532 nm, and at 532 nm measurements were additionally
made for particles exposed to elevated RH (~85%). As with the UCD PAS, the CRDS operated behind a
thermodenuder on a 2.5 or 5 minute cycle. PSAP and nephelometer measurements were made over the
period 3-28 June, 2010, with a data gap from the period 16-20 June. The CRD measurements and PAS
measurements at 405 nm were made starting 20:00 PDT on 16 June through 09:00 PDT on 29 June. Due
to a laser malfunction, the PAS measurements at 532 nm started at 12:00 PDT on 19 June. The
thermodenuder measurements began at 12:00 PDT on 20 June.

Particle mobility diameter ($d_{p,m}$) size distributions from 12.2 to 736.5 nm were measured using a

scanning mobility particle sizer (SMPS). Particle aerodynamic diameter ($d_{p,a}$) size distributions from 542
nm to 19,810 nm were measured using an aerodynamic particle sizer (APS). The APS size distributions
were converted to mobility-equivalent size distributions, assuming spherical particles and a particle
material density of 2.0 g cm$^{-3}$ and accounting for the Cunningham slip correction. The use of a material
density of 2.0 g cm$^{-3}$ implicitly assumes that the larger particles characterized by the APS were primarily
dust or sea spray. The resulting APS distribution was merged with the SMPS distribution to generate a
time-series of the mobility size distributions with sizes over the entire range (12.2-19,810 nm). Because
the $PM_1$, $PM_{2.5}$ and $PM_{10}$ designations are associated with aerodynamic diameters, mobility equivalent
cut-diameters must be determined. The mobility equivalent cut-diameters are (assuming a density of 2 g
cm$^{-3}$) 700 nm, 1750 nm and 7200 nm, respectively. (For simplicity, we will continue to refer to the sub
and supermicron particle ranges based on the aerodynamic size.) The merged size distribution was
truncated to an upper limit of 7200 nm, and the campaign average for the T0 site is shown for reference
in Figure S2. One complication at the T0 site is that the APS did not collect data from 13:30 PDT on June
22 onwards, and thus information about the supermicron particle size distribution and mass concentrations
are not available after this date.

Particle composition was monitored at T0 using the single particle laser ablation time-of-flight mass

spectrometer (SPLAT-II) from 3-28 June (Zelenyuk et al., 2009). SPLAT-II characterizes the composition



of individual particles and can be used to build a statistical picture of the distribution of particle types, as
defined by the uniqueness of and similarities between their mass spectra (Zelenyuk et al., 2008). Analysis
of the single particle mass spectra from SPLAT-II indicate a diversity of particle types, including dust,
sea salt-containing (SS), combustion derived (including particles categorized as soot, biomass burning
(BB), primary organic (POA) and hydrocarbon (HC)), amine-containing, and mixed sulfate/organic.
SPLAT-II samples particles between ~50 nm and 2 μm, although the sampling efficiency varies with
particle size and with particle shape. SPLAT-II is optimized for particles with vacuum aerodynamic
diameters ($d_{va}$) between 100 and 600 nm, and detects larger particles with reduced relative efficiency.
Non-refractory submicron particle matter (NR-PM) composition was measured using an Aerodyne
high resolution time-of-flight aerosol mass spectrometer (HR-AMS) (Canagaratna et al., 2007). The NR-
PM components measured include organics, sulfate, nitrate, ammonium and chloride. Data during the first
week of the campaign (June 3-12) are especially noisy due to instrumental problems. The AMS data were
processed and NR-PM concentrations determined using standard methods, assuming a collection
efficiency of 50%. Positive matrix factorization was applied to the dataset (Zhang et al., 2011) and three
factors associated with the organic aerosol (OA) were determined. One of these was characterized as a
more-oxidized OA factor (OOA) while two were characterized as less-oxidized OA factors, which will
be referred to here as the hydrocarbon-like OA factor (HOA) and were most likely cooking- and traffic-
related (Atkinson et al., 2015). Black carbon mass concentrations were measured with a single particle
soot photometer (SP2; DMT) (Schwarz et al., 2006). The SP2 was calibrated using mobility size selected
Aquadag particles, using a size-dependent particle density (Gysel et al., 2011). The reported
concentrations have been multiplied by a factor of 1.53 to account for the difference in sensitivity of the
SP2 to Aquadag compared with fullerene soot (R. Subramanian, Personal Communication), which is
thought to be a reasonable proxy for diesel soot (Laborde et al., 2012a). The CARES SP2 instruments
measured BC-containing particles with volume equivalent core diameters between 30 and 400 nm,
although quantification below $d_{p,ved} < \sim 100$ nm becomes generally more challenging and can vary between
instruments (Laborde et al., 2012b). No adjustment of the reported concentrations for black carbon
containing particles outside of the SP2 detection size range has been made, thus the reported
concentrations may be underestimated (Cappa et al., 2014).
Gas-phase concentrations of the sum of NO and $NO_2$ (= $NO_x$) and the sum of nitrogen oxides (=
$NO_y$) were measured using a 2-channel chemiluminescence instrument (Air Quality Design, Inc.) in which
$NO_2$ is photolyzed to NO and $NO_y$ is converted to NO on a Mo catalyst. Gas-phase concentrations of
hydrocarbons, in particular of toluene and benzene, were measured using a proton transfer reaction mass



spectrometer (PTR-MS). These measurements can be used to estimate the average photochemical age
($PCA$) of the air mass (Roberts et al., 1984), with:

$$PCA_{NOx} = -\frac{1}{k_{rxn}[OH]} \ln\left(\frac{[NO_x]}{[NO_y]}\right)$$                          (1)
where $k_{rxn}$ is the reaction rate coefficient for the OH + NO$_2$ reaction (7.9 x 10$^{-12}$ cm$^3$ molecules$^{-1}$ s$^{-1}$;
(Brown et al., 1999)), and

$$PCA_{HC} = \frac{\ln(ER) - \ln\left(\frac{[benzene]}{[toluene]}\right)}{(k_b - k_t)[OH]}$$                          (2)

where $ER$ is the emission ratio between benzene and toluene, assumed here to be 3.2 (Warneke et al.,
2007), and $k_b$ and $k_t$ are the reaction rate coefficients for reaction of benzene (1 x 10$^{-12}$ cm$^3$ molecule$^{-1}$ s$^{-1}$)
and toluene (5.7 x 10$^{-12}$ cm$^3$ molecule$^{-1}$ s$^{-1}$) with OH, respectively (Atkinson et al., 2006). Although there
are challenges in interpreting $PCA$ estimates quantitatively due to e.g. mixing of different sources (Parrish
et al., 2007) and weekend/weekday differences (Warneke et al., 2013), $PCA$ nonetheless provides a useful
estimate of the extent of photochemical processing.
### 2.2. Measurements at the T1 Site

A similar suite of measurements were made at the T1 site as at the T0 site, including light absorption

at 470, 532, 660 nm by PSAP, light scattering at 450, 550, 700 nm by nephelometer, particle size by
SMPS and APS and submicron particle composition by HR-AMS and SP2. The SMPS deployed at T1
measures particle number distribution in the range of 10 – 858 nm in $d_{p,m}$. Analysis of the HR-AMS data
using positive matrix factorization identified two distinct OOA factors, one of which was mainly
associated with biogenic emissions and the other representative of secondary organic aerosol (SOA)
formed in photochemically processed urban emissions. HOA was also observed at T1 but it on average
accounted for only ~10% of the OA mass. Details on HR-AMS and SMPS measurements at T1 and
associated data analysis are given in (Setyan et al., 2012; Setyan et al., 2014). The particle scattering and
absorption measurements were made nearly continuously from 3-28 June, 2010. Light absorption
measurements were also made using different PAS instruments, although these are not utilized here.
PTR-MS measurements of toluene and benzene are available from 3-28 June, 2010. Although NO and




NO$_y$ measurements were made, NO$_x$ was not measured. Thus, it is only possible to estimate *PCA* at the
T1 site using the benzene-toluene method (Equation 2).

One additional way in which particle composition was characterized at T1 was with a particle

ablation laser-desorption mass spectrometer (PALMS) (Cziczo et al., 2006). The PALMS is similar to the
SPLAT-II in that single particle mass spectra are collected for particles between about $d_{p,a}$ 150 nm and 2
μm, which are used to build a statistical picture of particle types. Analysis of the single particle mass
spectra from PALMS at the T1 site yielded the following particle types: dust (termed MinMet for
mineralogical/meteoric), sea salt-containing (SS), combustion derived (including particles categorized as
soot, biomass burning (BB), or oil), mixed sulfate/organic and "other." Results from the PALMS
measurements have been previously reported in Zaveri et al. (2012).

### 2.3 Derived particle properties

Using the alternating (i.e. sequential) PM$_1$ and PM$_{10}$ measurements, the properties of supermicron

particles specifically have been estimated from the difference between PM$_{10}$ and PM$_1$, with
$$b_{x,super} = \frac{b_{x,PM10}(t-1) + b_{x,PM10}(t+1)}{2} - b_{x,PM1}(t) \tag{3}$$
where *x* indicates absorption or scattering and where the *t* values indicate the average concentration over
each 6 min averaging period (i.e. the PM$_1$-PM$_{10}$ cycle time). The fraction of absorption or scattering from
PM$_1$ or supermicron PM is therefore defined as:
$$f_{x,PM1} = \frac{b_{x,PM1}}{b_{x,PM10}} \tag{4a}$$
$$f_{x,super} = \frac{b_{x,super}}{b_{x,PM10}} \tag{4b}$$
where *x* again indicates absorption or scattering. These ratios give an indication of the contribution of
submicron or supermicron particles to the total absorption or scattering, e.g. larger values of $f_{sca,PM1}$
indicate greater dominance of the submicron particle mode in terms of total scattering.

Light absorption measurements are used to determine the absorption Ångstrom exponent (*AAE*),

which characterizes the wavelength dependence of absorption and is given as:
$$AAE_{\lambda1,\lambda2} = -\frac{\log\left(\frac{b_{abs,\lambda1}}{b_{abs,\lambda2}}\right)}{\log\left(\frac{\lambda1}{\lambda2}\right)} \tag{5}$$



where λ1 and λ2 indicate different wavelengths. It is often assumed that "pure" black carbon (BC)
particles have an *AAE* close to 1 and that values >1 indicate the presence of light absorbing organics
(referred to as brown carbon, or BrC) or dust, which tend to exhibit absorption that increases sharply as
wavelength decreases. The *AAE* is dependent upon the chosen wavelength pair. The specific wavelength
pair used to calculate *AAE* will be indicated using the notation $AAE_{\lambda1-\lambda2}$. Related, the difference in the
$PM_{10}$ and $PM_1$ AAE can be calculated:
$\Delta AAE_{10-1} = AAE_{PM10} - AAE_{PM1}$ (6)
The scattering Ångstrom exponent (*SAE*) is also commonly used to characterize the relative
contributions from sub- and supermicron particles, and is defined analogously to the *AAE* as:
$SAE_{\lambda1,\lambda2} = -\dfrac{\log\left(\frac{b_{sca,\lambda1}}{b_{sca,\lambda2}}\right)}{\log\left(\frac{\lambda1}{\lambda2}\right)}$ (7)
Larger values of the *SAE* correspond to overall smaller particles, and have been calculated for $PM_1$, $PM_{10}$
and $PM_{super}$. A similar parameter, the extinction Ångstrom exponent, *EAE*, can be calculated for $PM_{2.5}$
using the measured $b_{ext}$.
The fraction of extinction due to scattering is characterized through the single scatter albedo (SSA),
which can be written in multiple ways depending on whether $b_{ext}$, $b_{abs}$ or $b_{sca}$ were the directly measured
properties:
$SSA = \dfrac{b_{ext}-b_{abs}}{b_{ext}} = \dfrac{b_{sca}}{b_{ext}} = \dfrac{b_{sca}}{b_{sca}+b_{abs}}$ (8)
The angular dependence of scattering is characterized through measurement of the backscatter
coefficients, $b_{bsca}$. The fraction of light that is backscattered, relative to the total scattering, is calculated
as
$f_{bsca} = \dfrac{b_{bsca}}{b_{sca}}$ (9)
The backscatter fraction is an important climate-relevant parameter as particle radiative effects depend in
part on the extent to which incoming solar radiation is reflected back to space versus absorbed within the
Earth system. The backscatter fraction is commonly converted to an asymmetry parameter, $g_{sca}$, and the
empirically derived relationship between these is (Andrews et al., 2006):
$g_{sca} = -7.143889 \cdot f_{bsca}^3 + 7.464439 \cdot f_{bsca}^2 - 3.9356 \cdot f_{bsca} + 0.9893$ (10)





The asymmetry parameter is the intensity-weighted average of the cosine of the scattering angle and
ranges from -1 (all backscatter) to 1 (all forward scatter).
Using the measurements made behind the thermodenuder at the T0 site (i.e. the PAS and CRD
measurements), various ratios and differences can be determined. The ratio between the denuded and
undenuded extinction (i.e. the fraction of extinction remaining for $PM_{2.5}$) provides a measure of the
particle volatility, with smaller values indicating more volatile particles:
$$f_{ext,TD} = \frac{b_{ext,TD}}{b_{ext,amb}} = \frac{b_{ext}(t-1)+b_{ext}(t+1)}{2b_{ext}(t)}$$    (11)
where *TD* indicates the thermodenuded and *amb* indicates the ambient time periods, and the 2$^{nd}$ equality
shows how the sequential TD and ambient measurements were accounted for similar to Eqn. 3. The
change in particle hygroscopicity upon thermodenuding is calculated as
$$\Delta\gamma_{RH} = \gamma_{RH}(amb) - \gamma_{RH}(TD)$$    (12)
where
$$\gamma_{RH} = \log\left(\frac{b_{ext,high}}{b_{ext,low}}\right) / \log\left(\frac{100-RH_{low}}{100-RH_{high}}\right)$$    (13)
and the *high* and *low* refer to the humidified and dried CRD measurements and again accounting for the
sequential nature of the *TD* and *amb* measurements. The parameter $\gamma_{RH}$ can be thought of as the optical
hygroscopicity (i.e. a measure of the affinity of particles towards water), although it does have some
dependence on particle size and thus there is not a 1-to-1 relationship between $\gamma_{RH}$ and particle
hygroscopicity(Atkinson et al., 2015). In general, for a given amount of particle growth due to water
uptake, $\gamma_{RH}$ is larger for smaller particles.
The difference in the AAE between the ambient and thermodenuded states can also be determined:
$$\Delta AAE_{amb-TD} = AAE_{amb} - AAE_{TD}$$    (14)
Mass absorption, scattering and extinction coefficients (*MAC*, *MSC* and *MEC*, respectively) have been
calculated for the various wavelengths and PM size ranges. Using scattering as an example,
$$MSC_X \left(\frac{m^2}{g}\right) = \frac{b_{sca}}{[X]}$$    (15)
where [X] is the mass concentration of the reference species of interest, such as BC or the total PM. In
the case of [BC], the SP2 measurements are used. For total PM, the integrated volume concentrations





were used to estimate [PM], assuming spherical particles. For supermicron particles it was assumed that
$\rho_p$ = 2.0 g cm$^{-3}$ and for submicron particles the density was 1.3 g cm$^{-3}$ (Setyan et al., 2012).
The size distributions have been used to calculate median surface-weighted particle diameters
($d_{p,surf}$) as:
$$d_{p,surf} = \frac{\int_{d_{p,low}}^{d_{p,high}} d_p \cdot \frac{dN}{dlog\, d_p} dlog\, d_p}{\int_{d_{p,low}}^{d_{p,high}} \frac{dN}{dlog\, d_p} dlog\, d_p}$$
(16)

where dN/dlog$d_p$ is the observed number-weighted size distribution over the size range of interest ($d_{p,low}$
to $d_{p,high}$). In this study, $d_{p,surf}$ values have been separately calculated for the total PM$_{10}$ distribution and
for the supermicron particle range.
## 3. Results and Discussion
### *3.1 Size dependence of optical properties*
#### *3.1.1 Light scattering*
Supermicron particles contributed substantially to the dry particle scattering at both T0 and T1
(Figure 1). The average $f_{sca,PM1}$(550 nm) at T0 was 0.48 ± 0.17 (1$\sigma$) and at T1 was 0.55 ± 0.16 (1$\sigma$). (If
the time period where the scattering measurements were not available at T0 is excluded from the T1
average, the average $f_{sca,PM1}$ is 0.57 ± 0.15 (1$\sigma$).) Thus, nearly 50% of the dry scattering at these two sites
was, on average, due to supermicron particles during the campaign period. This observation is consistent
with results from Kassianov et al. (2012), who calculated scattering and direct radiative forcing for these
two sites based on observed size distributions and concluded that supermicron particles contribute
substantially to the total scattering. Because scattering generally increases more rapidly with decreasing
wavelength for small particles, the $f_{sca,PM1}$ is larger for 450 nm (= 0.59 at T0 and 0.67 at T1) and smaller
for 700 nm (= 0.34 at T0 and 0.41 at T1) compared to 550 nm. The similarity of the $f_{sca,PM1}$ values between
the two sites is noteworthy given that the T0 site is situated much closer to urban Sacramento than is the
T1 site. The $f_{sca,PM1}$ values at the two sites show a strong, linear correlation with the $SAE_{450,550}$ (Figure
2a,b), which is not entirely surprising since the $SAE$ provides an indication of the mean, optically-
weighted particle size. There is a similarly strong relationship with the $SAE$ values calculated from the
other wavelength pairs (Figure S3 and Figure S4 show results for the T0 and T1 sites, respectively, for
comparison). The best fit from a one-sided linear fit to $SAE_{450,550}$ versus $f_{sca,PM1}$ at T0 is $SAE_{450,550}$ = 2.69





$f_{\text{sca},550,\text{PM1}}$ - 0.05 ($r^2 = 0.88$) and at T1 is $SAE_{450,550} = 2.66\, f_{\text{sca},550,\text{PM1}} + 0.06$ ($r^2 = 0.91$). (Values for other
combinations of wavelengths for T0 and T1 are reported in Table S1.)

Although the *SAE* values exhibit a linear relationship with $f_{\text{sca},\text{PM1}}$, they exhibit a more complex

relationship with the median surface-area weighted particle diameter of the entire distribution ($d_{\text{p,surf,PM10}}$).
Although the *SAE* values generally decrease with increasing $d_{\text{p,surf,PM10}}$, as one might expect since an
increase in $d_{\text{p,surf,PM10}}$ presumably corresponds to an increase in the supermicron fraction of scattering,
there is much greater scatter compared to the clear relationship with $f_{\text{sca},\text{PM1}}$, and clear periods when a
monotonic relationship is not observed (Figure 2c,d). The derived $d_{\text{p,surf,PM10}}$ values are sensitive to the
exact shapes of the sub- and supermicron modes, and the *MSC* for supermicron particles, in particular, is
also sensitive to the shape of the supermicron size distribution (discussed further below in Section 3.1.2).
Consequently, there is not a straight-forward relationship between the *SAE* and $d_{\text{p,surf,PM10}}$ and the *SAE*
cannot be used to predict average properties of the overall sub- plus supermicron size distribution.
However, the strong, linear relationship between the *SAE* and $f_{\text{sca},\text{PM1}}$ and the close correspondence
between the two sites, despite apparent variations in the underlying size distributions, suggests that the
relationships derived here may be sufficiently general to be applied in other locations, although this needs
to be verified. This in turn indicates that observations of *SAE* can be used to quantitatively estimate the
fractional contribution of sub and supermicron particles to the total scattering with an uncertainty of ~15%
based on the spread in the data. Thus, the relationships derived here may be useful for application or
comparison with remote sensing methods, such as the AERONET sun photometer network (Schuster et
al., 2006) or satellites (Ginoux et al., 2012).

One factor that can influence the sub- versus supermicron contributions to the total scattering is the

extent of photochemical processing. Photochemical processing leads to the production of condensable
material that will tend to condense according to the particle Fuchs-corrected surface area. As such,
photochemical processing and secondary PM formation, especially SOA, will lead to preferential growth
of the submicron mode diameters (which grow more for a given amount of material condensed) and will
lead to an increase in the submicron scattering in particular. The $f_{\text{sca},\text{PM1}}$ at T0 exhibits a general increase
with PCA (characterized by the [NO$_\text{x}$]/[NO$_\text{y}$] and/or [benzene]/[toluene] ratios), although there is a fair
amount of scatter at low photochemical age (Figure 3a,b). The $f_{\text{sca},\text{PM1}}$ at T1 shows completely different
behavior, with $f_{\text{sca},\text{PM1}}$ exhibiting very little dependence on PCA, although there is possibly a slight
decrease (Figure 3c). There is generally good correspondence between the [NO$_\text{x}$]/[NO$_\text{y}$] and
[benzene]/[toluene] ratios measured at T0, indicating that the different results at T1 are unlikely to result
from use of a particular PCA marker (Figure 3d). The PCA at T1 is on average much larger than at T0.





The PCA at the T0 site exhibits a clear peak around 15:00 PDT (Figure 3e). The PCA diel profile at T1
is comparably much broader and exhibits a less-pronounced peak, but that also occurs in the late afternoon
(around 16:00 PDT) (Figure 3e). This broadening and different temporal dependence likely reflects the
downwind location of the T1 site and the general flow patterns in this region (Fast et al., 2012). It seems
likely that the difference in variation of the $f_{sca,PM1}$ with PCA between the two sites is related to these
difference. At T0, the measured $b_{sca,PM1}$ may reflect a relatively local production of submicron particulate
mass whereas at T1 the $b_{sca,PM1}$ is more reflective of regional conditions. The $b_{sca,super}$ at both sites will
have some regional contribution (in particular sea spray), but also a strong local contribution. Wind speeds
were typically largest in the mid- to late afternoon at both sites, although overall the diel behavior was
much clearer at T1 than at T0 and with a larger amplitude (Figure 3e). However, the wind speeds were on
average larger at T0. Thus, it seems reasonable to conclude that local emission of supermicron
particulates, possibly re-suspended road dust or from agricultural sources, in the afternoon at T1
counteracts the influence of growth of the regional submicron particulates, leading $f_{sca,PM1}$ to be nearly
independent of PCA at this site. In contrast, at T0 the local photochemical production of new submicron
PM mass appears to be sufficiently strong to lead to an increase in $f_{sca,PM1}$ with PCA.
*3.1.2 Light absorption*
Light absorption at both sites was dominated by submicron particles, although a small fraction may
also be from supermicron particles (Figure 1). The average $f_{abs,PM1}$ at 532 nm at T0 was $0.89 \pm 0.14$ (1σ)
and at T1 was $0.85 \pm 0.17$ (1σ). At T1 there is a slight indication that $f_{abs,super}$ decreased as $f_{sca,super}$
decreased, but no such relationship is clearly evident at T0 (Figure 1g-h). Such potential relationships
must be viewed with some amount of caution, as the PSAP requires correction for particle scattering and
the extent of forward versus backward scattering is particle size dependent. It is also known that the PSAP
shows an additional sensitivity to particle size due to differences in the depth of penetration of particles
into the filter (Nakayama et al., 2010), which might influence the measurements here.
That light absorption is dominated by submicron particles suggests that black carbon, and possibly
brown carbon, make up the majority of the light absorbing particle burden. The $AAE_{PM1}$ values at both T0
and T1 exhibit reasonably normal distributions (Figure 4). The spread at T0 was substantially smaller than
at T1. The average $AAE_{PM1}$ values are slightly larger than unity at both sites ($1.21 \pm 0.13$ and $1.33 \pm 0.22$
for T0 and T1, respectively, where uncertainties are 1 standard deviation; Table 2), and do not show any
pronounced dependence on the wavelength pair chosen. Black carbon is typically thought to have an $AAE$
close to unity (Cross et al., 2010). The average $PM_1$ $AAE_{PM1,450-532}$ are identical between the sites, whereas
the $AAE_{PM1,450-660}$ and $AAE_{PM1,532-660}$ are slightly larger at T1. The larger spread in the $AAE_{PM1}$ values for





all wavelength pairs at T1 suggests that the two PSAP instruments were not entirely identical and had
somewhat different noise characteristics, making it difficult to establish whether these small differences
are real. One method that has been used to estimate the contribution of brown carbon relative to black
carbon is to extrapolate the observed $b_{abs}$ at longer $\lambda$ (e.g. 660 nm) to shorter wavelengths assuming that
$AAE = 1$ and that absorption by brown carbon at long $\lambda$ is zero (Gyawali et al., 2009; Yang et al., 2009;
Chung et al., 2012; Lack and Langridge, 2013). To the extent that this method is appropriate, and some
have argued it may not be (Lack and Langridge, 2013), it provides an estimate of $b_{abs}$ for pure BC
(assuming that $AAE_{BC} = 1$ exactly) and the contribution from brown carbon can then be estimated by
subtracting the pure BC $b_{abs}$ from the total. Given the observed $AAE_{PM1}$ values, this spectral differencing
method suggests that brown carbon contributes ~6% at T0 and up to 11% at T1 to submicron particle light
absorption at 450 nm; if the actual $AAE_{BC}$ were >1, as possibly suggested by the $AAE_{532-660}$ measurements
at both sites, then the attributed brown carbon fraction would be even smaller. These relatively minor
contributions from brown carbon are consistent with the conclusions of Cappa et al. (2012) and indicate
that in this region the summertime submicron particulate light absorption is predominately from black
carbon.
There are two likely origins of the supermicron absorption: penetration of BC into the supermicron
size range, likely from coagulation of BC with larger particles or tailing of the predominately submicron
BC size distribution, or dust (assuming the observed supermicron absorption is not simply an experimental
artifact). Dust is known to contribute substantially to the $PM_{10}$ burden in Sacramento, with sources
including roadways, agricultural activity and disturbed open residential areas (California Air Resources
Board, 2005; http://www.arb.ca.gov/pm/pmmeasures/pmch05/pmch05.htm), as well as long range
transport (Ewing et al., 2010). The single particle measurements from the SPLAT-II instrument at T0
indicate that both BC-containing and dust particles are observed in the supermicron size range, along with
substantial contributions from sea salt-containing particles that are likely of marine origin (Figure S5).
The PALMS instrument detected similar particle types at the T1 site (Zaveri et al., 2012) Although
informative, these measurements unfortunately cannot be used to quantitatively assess the relative
contributions of the different absorbing particle types to the supermicron absorption because both
instruments sample only over a subset of the entire supermicron size range, e.g. the SPLAT-II only up to
$d_{v,a} \sim 2$ μm. Nonetheless, the single particle composition measurements provide support for the likely
origin of the supermicron absorption being from either BC penetration or dust.
Dust and BC should be distinguishable based on the observed spectral properties and chemical
composition. Although the optical properties of dust vary by location and source, dust is generally thought





to have $AAE$ values > 1, with typical reported values of ~1.5-3 (Yang et al., 2009; Russell et al., 2010;
Bahadur et al., 2012), larger than is typically observed for black carbon. The measured average $AAE_{super}$
values were typically greater than unity and larger than the $AAE_{PM1}$ (Table 2), suggestive of a dust
influence. However, there are two important points to consider. First, although the $AAE_{super}$ values were
approximately normally distributed, the distributions were substantially broader than the distributions for
the submicron particles, with a range of about $0.5 < AAE_{super} < 3$ (Figure 4). Second, the $AAE_{super}$ values
exhibited a notable wavelength-pair dependence, with the largest values observed for the 450-532 nm pair
and the smallest for the 532-660 nm pair and where the wavelength-dependence at T0 was much larger
than at T1 (Table 2;Figure 4). The large spread in the $AAE_{super}$ values may reflect the small magnitudes of
the absolute $b_{abs,super}$ values and the use of the difference method to determine the $b_{abs,super}$ (i.e. noise), but
could also indicate a greater diversity in $AAE_{super}$ values compared to $AAE_{PM1}$ due to, perhaps, time-
varying contributions from dust and BC. As a test, if the averaged $AAE_{PM1}$ values are restricted to periods
when the absolute absorption was relatively low ($< 0.6$ Mm$^{-1}$), but still generally larger than the $b_{abs,super}$,
there is no substantial additional broadening of the distribution. This suggests that the broadening of
$AAE_{super}$ may be real and that the latter interpretation—diversity in individual $AAE_{super}$ values—may be
appropriate. However, since there were insufficient periods where the $b_{abs,PM1}$ values were as low as the
$b_{abs,super}$, changes in the shape of the $AAE_{PM1}$ distribution cannot be assessed under the exact same
conditions, and thus the possibility that the greater scatter simply reflects the low $b_{abs,PM10}$ values cannot
be ruled out.  (The 0.6 Mm$^{-1}$ threshold was chosen to allow for a sufficient number of $AAE_{PM1}$ values to
remain to be used to develop a histogram.) Also, if one considers the relationship between $AAE_{PM10}$ and
the $f_{sca,PM1}$, there is no obvious broadening of the $AAE$ distribution at smaller values of $f_{sca,PM1}$. There is
also some indication of correlations between both $b_{abs,super}$ and $b_{abs,PM1}$ and between $b_{abs,super}$ and [BC] (as
measured by the SP2) (Figure 5). Although the correlation coefficients are relatively small ($r^2 = 0.45$ and
0.25 at T0 and T1, respectively), this could indicate contributions from penetration of BC into the
supermicron mode, which could help to explain why most of the $AAE_{super}$ values are smaller than is typical
for pure dust but larger than for BC.

Further insight into the nature of the supermicron particles comes from consideration of the $MAC$

and $MSC$ values, which are intensive properties like the $AAE$. The $MAC$ and $MSC$ for supermicron
particles have been assessed by comparing $b_{abs,super}$ and $b_{sca,super}$ with the supermicron mass concentration
([PM$_{super}$]) as estimated from the measured size distributions (Figure 6a-b). The [PM$_{super}$] values were
estimated assuming a density of 2 g cm$^{-3}$ and spherical particles. In theory, the $MSC$ is size dependent,
falling off rapidly from ~4 m$^2$ g$^{-1}$ to ~1.5 m$^2$ g$^{-1}$ in going from $d_{p,m} = 700$ to 1000 nm (with $d_{p,m} = 700$ nm
corresponding approximately to $d_{p,a} = 1000$ nm when density = 2 g cm$^{-3}$) and ranging from ~ 0.5 m$^2$ g$^{-1}$





to 1.5 $m^2$ $g^{-1}$ for larger particles (see Figure S6 and (Clarke et al., 2004)). Thus, smaller *MSC* values
generally correspond to overall larger particles. The observed $MSC_{super}$ exhibit an inverse relationship
with $d_{p,surf}$ for the supermicron particles (Figure 6e-f). To our knowledge, this is the first explicit
demonstration of the theoretically expected inverse relationship from ambient observations. These
observations indicate that the nature of the supermicron particle size distributions do vary with time, with
some time periods containing larger supermicron particles and some smaller. This variability in size
suggests variations in the supermicron particle sources, and consequently in the chemical nature of the
particles, discussed further below.

The relationship between $b_{abs,super}$ and [$PM_{super}$] exhibits a great deal of scatter (Figure 6c-d), most

likely due to the small values of $b_{abs,super}$ and to the variability in the $PM_{super}$ sources, including particle
density and size. The *MAC* values at 532 nm for the supermicron particles range from ~0 to ~0.06 $m^2$ $g^{-1}$
at both sites. Further, the $MAC_{super}$ values exhibit a notable dependence on the $d_{p,surf}$ (Figure 6g-h). In
general, when $d_{p,surf}$ is on the small side (~2 μm) the $MAC_{super}$ is very close to zero and it tends to increase
with $d_{p,surf}$. Apparently, the particles from sources that produced, on average, smaller supermicron
particles were less absorbing than the particles from sources that produced larger particles. A plausible
explanation is that the larger particles are dust while the smaller (yet still supermicron) particles are a
combination of non-absorbing sea spray particles and other particle types that are penetrating from the
submicron mode. This hypothesis is generally supported by examination of HYSPLIT back trajectories
(Draxler and Rolph) for each day of the campaign (Figure S7), as well as by comparison with the source-
region identification in Fast et al. (Fast et al., 2012), with smaller $d_{p,surf}$ values for the supermicron particles
generally corresponding to periods when the air masses were identified as containing a greater "Bay Area"
contribution. Considering three specific days as examples, two (11 and 16 June) when the $MSC_{super}$ were
particularly small (corresponding to larger particles) and one (15 June) when the $MSC_{super}$ were larger,
clear differences in the air mass origins can be seen. Specifically, the back trajectories on 11 and 16 June
indicate that the air mass came from almost due north, consistent with a terrestrial origin for the particles
while the back trajectory on 15 June indicates that the air mass had passed over the San Francisco Bay
Area and before that came from along the CA coast. These back trajectories are generally consistent with
the idea that when the overall size distribution is skewed towards smaller supermicron particles (smaller
$d_{p,surf}$ and larger $MSC_{super}$) the air masses are more impacted by sea spray particles, while when the size
distribution is skewed towards larger particles there is a greater relative dust contribution.

Even though the $MAC_{super}$ exhibits a pronounced relationship with $d_{p,surf}$, there is actually minimal

dependence of $b_{abs,super}$ on $d_{p,surf}$ (Figure S8). There is, however, a relatively strong relationship between



[$PM_{super}$] and $d_{p,surf}$, with larger [$PM_{super}$] usually corresponding to smaller $d_{p,surf}$ (Figure S8). This suggests
that the small $MAC_{super}$ values at small $d_{p,surf}$ result from substantial inputs of non-absorbing supermicron
particles, which does not necessarily alter the observed $b_{abs,super}$ but does serve to increase the [$PM_{super}$],
thereby depressing the $MAC_{super}$ values. The $MAC_{super}$ is approximately 0.06 m$^2$ g$^{-1}$ when $d_{p,surf}$ is large
(i.e. >3.5 μm; Figure 6). If it is assumed that the major contributor to supermicron absorption when $d_{p,surf}$
is large is dust then a value for the imaginary refractive index ($k$) for dust in this region is estimated from
Mie theory. Assuming spherical particles with $d_p$ = 3.5 μm with density = 2 g cm$^{-3}$ and a real refractive
index of either 1.5 or 1.6, the $k$ is ~0.0035$i$. However, this estimate assumes that all of the $PM_{super}$ mass
is dust and that no other absorbing species contribute. If some of the $PM_{super}$ mass is attributed to non-
dust species, then the derived dust-specific $MAC$ and $k$ would be larger. Alternatively, if BC contributes
substantially to the observed supermicron absorption, which seems likely, then the dust-specific $MAC$ and
$k$ would be smaller. Most likely, the above values are upper-limits. Despite these uncertainties, the
observed $MAC$ and $k$ are similar to reported estimates for dust in the Xianghe area in China, where $MAC_{dust}$
= 0.048 m$^2$ g$^{-1}$ at 520 nm and where the reported $MAC$ has been adjusted to a density of 2.0 g cm$^{-3}$ (Yang
et al., 2009). Overall, although the contribution of supermicron particles to the total absorption is small in
this region, it nonetheless must be considered.
*3.1.3 Relationship between scattering and absorption*
There has been increasing interest in the relationship between the absorption Ångstrom exponent
and the scattering Ångstrom exponent (Yang et al., 2009; Russell et al., 2010; Bahadur et al., 2012; Giles
et al., 2012; Cazorla et al., 2013; Costabile et al., 2013). The wide range of $SAE$ values observed here
allows for assessment of the $AAE$ vs. $SAE$ relationship in a constrained environment. The observed
$AAE_{532-660}$ vs. $SAE_{450-550}$ relationships for PM$_{10}$, submicron and supermicron particles are shown in Figure
7a-c. The observed $AAE$ values at both sites fall in a fairly narrow range centered around 1.2 for PM$_{10}$ and
submicron particles, with much greater scatter for supermicron particles, consistent with Figure 4. The
submicron $SAE$ values are >1.8 and the supermicron $SAE$ values are generally <0.2, while the PM$_{10}$ $SAE$
values span the range 0.3 – 2. The wide range of $SAE$ values for PM$_{10}$ results from time-varying
contributions of supermicron and submicron particles to the total scattering.
Cazorla et al. (2013) previously proposed a classification scheme based on the position in the $AAE$
vs. $SAE$ space (c.f. their Figure 1). They classified particles with $AAE$ < 1 and $SAE$ > 1.5 as "EC
dominated," and where EC stands for elemental carbon (which is approximately equivalent to BC
(Andreae and Gelencser, 2006; Lack et al., 2014)). Here, almost none of the observations fall in this space,
despite the submicron absorption being dominated by black carbon. Instead, the submicron measurements



fall primarily in the space encompassed by $1 < AAE < 1.5$ and $SAE > 2$, which Cazorla et al. (2013)
classified as an "EC/OC mixture" and where an implicit assumption was that the OC (organic carbon)
was absorbing in nature (i.e. BrC), thus leading to the elevated $AAE$ values compared to the "EC
dominated" region. These *in situ* measurements therefore suggest that the "EC/OC mixture" region should
better be classified as "EC dominated" (or equivalently "BC dominated"). These measurements indicate
that BrC contributions to submicron absorption can only be clearly identified if the $AAE$ is well-above
1.5. Given that almost none of the submicron $AAE$ values were $< 1$, the suggestion by Bahadur et al.
(2012) that a "low-end baseline" $AAE$ value of 0.55 ($\pm$ 0.24) that is related to "pure EC" seems unlikely
to be correct and is more likely a result of a subset of the data points considered in that study having large
uncertainties due to low signal levels. (Lower $AAE$ values can be obtained if a wavelength pair is selected
in which the wavelengths differ substantially and there is curvature in the $b_{abs}$ vs. wavelength relationship
(Bergstrom et al., 2007).) This conclusion is consistent with that of (Russell et al., 2010) and with the *in
situ* observations of (Yang et al., 2009) and the remote sensing observations of (Giles et al., 2012).
Cazorla et al. (2013) also classified particles having $1 < AAE < 1.5$ and $SAE < 1$ as being a "Dust/EC
mix", and those with $AAE < 1$ and $SAE < 1$ as being "Coated large particles." As the supermicron
contribution to scattering increases (and the $SAE$ decreases), the observed $AAE$ values, at T1 especially,
do not strongly deviate from the 1-1.5 range. The supermicron particles sampled here were a mixture of
sea spray and dust in varying amounts. This therefore suggests that the "Dust/EC mix" regime should be
reclassified to be more general, as it is not specific to "dust," only to "large particle/BC mixtures." The
measurements suggest that dust contributions can only be clearly elucidated when the $AAE > 1.5$, although
even when such large $AAE$ values are observed care must be taken if the absolute absorption values are
small (as is the case here for supermicron particles), corresponding to individual $AAE$ values with
substantial uncertainties. Similar caution is suggested for identification of particles in the "Coated large
particle" regime, as classified by Cazorla et al. (2013). The *in situ* measurements here suggest that
observations that fall within this regime are likely the result of measurement uncertainties due to low
signal levels, and do not correspond to the presence of "Coated large particles." Based on the observations
here, a new classification scheme using the $AAE$ and $SAE$ relationship is proposed (Figure 7d).
*3.1.3 Light backscattering*
The extent to which particles scatter light in the backward versus forward direction has an important
controlling influence on their climate impacts (Haywood and Shine, 1995). The backscatter fractions at
550 nm for $PM_{10}$, $f_{bsca,550,PM10}$, measured by the nephelometer ranged from 0.1 to 0.23, with an average
value of $0.137 \pm 0.024$ for T0 and $0.155 \pm 0.054$ for T1. These correspond to a $g_{sca}$ range for $PM_{10}$ of 0.40





535 to 0.67 and mean values of 0.57 ± 0.056 for T0 and 0.53 ± 0.054 for T1. This range of observed values is

536 comparable to measurements made at other locations (Andrews et al., 2006), but the averages are

537 somewhat smaller than $g_{sca}$ values calculated by *Kassianov et al.* (2012) at 500 nm for the T0 and T1 sites

538 (both $g_{sca,500} = 0.65$). The observed $g_{sca}$ versus $f_{sca,PM1}$ relationship is shown in Figure 8b,c for T0 and T1.

539 There is some general decrease in $g_{sca}$ when $f_{sca,PM1}$ increases at both sites, more clearly at T1 than at T0,

540 but at both sites there is substantial scatter in the data. Some of this scatter appears to be driven by

541 variations in the size of the submicron mode, as characterized by $d_{p,surf,PM1}$. In general, for a given $f_{sca,PM1}$

542 the observed $g_{sca}$ values are smaller when $d_{p,surf,PM1}$ is smaller.

543  This observed behavior is generally consistent with theoretical expectations. The theoretical

544 relationship between $g_{sca}$ and particle size for spherical particles is shown in Figure 8a. The calculated $g_{sca}$

545 increases nearly monotonically for diameters up to about 500 nm, reaching $g_{sca}$ ~0.75. In the supermicron

546 range above 1.5 μm the $g_{sca}$ is relatively constant around 0.75. In between 500 nm and 1.5 μm, the $g_{sca}$

547 exhibits a more complicated dependence on size. The steepness of the $g_{sca}$ versus $d_p$ relationship between

548 100 and 500 nm means that the observed $g_{sca}$ for $PM_{10}$ will be particularly sensitive to variations in the

549 submicron particle size distribution. However, the $g_{sca}$ will be less sensitive to variations in the

550 supermicron particle size distribution because the $d_p$ versus $g_{sca}$ relationship is generally flatter. Further,

551 we might expect some relationship between $g_{sca}$ and the fraction of scattering due to sub- or supermicron

552 particles to the extent that the two size regimes have generally distinct $g_{sca}$ values. Indeed, such behavior

553 is seen in the observations, in large part because the $d_{p,surf,PM1}$ values vary within the sensitive range (100-

554 500 nm).

555  Therefore, in an effort to account for this apparent co-dependence of $g_{sca}$ on $f_{sca,PM1}$ and $d_{p,surf,PM1}$,

556 the $f_{sca,PM1}$ values have been divided by the $d_{p,surf,PM1}$ values, with the ratio indicated as $R_g$. There is, in

557 general, a much stronger relationship between the $g_{sca}$ values and $R_g$ (Figure 8d,e) than there is with $f_{sca,PM1}$

558 alone, and much of the residual scatter seems to be driven by variations in the supermicron size

559 distribution. Linear fits give $g_{sca} = -38.5R_g + 0.66$ for T0 ($r^2 = 0.51$) and $g_{sca} = -51.5R_g + 0.66$ for T1 ($r^2 =$

560 0.71), and where $d_{p,surf,PM1}$ is in nm. Overall, the observations here demonstrate that the observable

561 backscatter coefficients depend importantly on the relative contributions of sub- versus supermicron

562 particles to the total scattering, but that the specific relationship between backscatter and the sub- or

563 supermicron scattering fraction is obscured by variations in the size distribution within each size range.

564 However, the greatest sensitivity of $g_{sca}$ is found for size variations within the submicron size range.





### *3.2 Influence of heating on optical properties*
At the T0 site, the UCD CRD and PAS instruments sampled alternately dried, ambient particles
(PM$_{2.5}$) or particles that had been passed through a thermodenuder (TD) that was held at 225 °C during
the study. As particles pass through the TD, some materials evaporate, including ammonium nitrate,
ammonium sulfate and many organics, while others do not, including black carbon, dust and sea salt. Loss
of these materials leads to changes in the optical properties, including the particle optically-weighted
hygroscopicity. The influence of heating on the optical properties is used here to further probe the particle
composition.
The observed fraction of extinction remaining after heating, $f_{ext,TD}$, for PM$_{2.5}$ varied from ~0.15 to
0.6, suggesting a wide range of particle volatility. This variability is strongly linked to the relative
contribution of sub- versus supermicron particles to the observed extinction; an approximately linear
relationship (with a positive slope) between $f_{ext,TD}$ and the *EAE* measured for the ambient particles was
observed (Figure 9). This increase in $f_{ext,TD}$ with decreasing *EAE* suggests that the supermicron
components are mostly non-volatile, consistent with a likely dust or sea salt contribution as identified
above. Further, this suggests that $f_{ext,TD}$ can be used as an indicator of particle size in the current study.
There is a cluster of points at the highest $f_{ext,TD}$ (the light green points in Figure 9) that were observed
during a specific overnight period when we suspect that the site was briefly impacted by large particles
produced as part of local road resurfacing.
The bulk particle hygroscopicity, characterized by $\gamma_{RH}$, did not vary monotonically with $f_{ext,TD}$
(Figure 10a). This is because the observed hygroscopicity depends on compositional variability within
both the sub- and supermicron modes (Atkinson et al., 2015). However, the change in the hygroscopicity
upon heating, $\Delta\gamma_{RH} = \gamma_{RH,ambient} - \gamma_{RH,TD}$, does exhibit a clear correlation with $f_{ext,TD}$, with larger $\Delta\gamma_{RH}$
corresponding to smaller $f_{ext,TD}$, i.e. for smaller, typically more volatile particles (Figure 10b). The
observed $\Delta\gamma_{RH}$ appear to cross over zero around $f_{ext,TD} = 0.4$. Apparently, for smaller particles that exhibit
greater overall mass loss upon heating, the $\gamma_{RH}$ tends to decrease with heating. This is as might be
expected, since one key residual component will be non-hygroscopic BC when the distribution is
dominated by smaller particles. However, when the distribution is dominated by larger particles,
evaporation leads to the residual particles appearing, on average, slightly more hygroscopic. This suggests
that the supermicron components that are susceptible to evaporation are lower-hygroscopicity material,
most likely organics but also, potentially, inorganics such as sulfate and nitrate, which have lower
hygroscopicity than sodium chloride. Some of the sea salt-containing particles observed during CARES
were found to be internally mixed with organics (likely organic acids) that displaced chloride (Laskin et



al., 2012), and organic salts are generally less hygroscopic than sea salt (Drozd et al., 2014). It is possible
that these organics evaporated in the TD, leaving behind more hygroscopic material, although such a
hypothesis requires further investigation.
The contribution to the total light absorption from non-BC materials that evaporate in the TD was
characterized by the absorption enhancement, $E_{abs}$, which is here taken as the ratio between the ambient
and thermodenuded $b_{abs}$. We have previously investigated the dependence of $E_{abs}$ on photochemical age
at CARES using the same data set as is being considered here, and separately the dependence on the
relative amount of "coating" (non-BC) material that is internally mixed with BC at the CalNex field study
(Cappa et al., 2012). It was found that the $E_{abs}$ increases by only a small amount as PCA and coating
amount increased. Here, we see that $E_{abs}$ exhibits some slight dependence on $f_{ext,TD}$, with somewhat larger
values observed at smaller $f_{ext,TD}$ (Figure 10c,d). It is difficult to establish whether this dependence
indicates that larger $E_{abs}$ would have been observed in Cappa et al. (2012) for the CARES dataset had
more material evaporated, but given that the $f_{ext,TD}$ here is determined predominately by changes in the
relative contributions from sub- and supermicron particles this seems unlikely. (The complementary
measurements from CalNex were for $PM_1$, not $PM_{2.5}$ as here, and thus the influence of supermicron
particles on the observations was substantially smaller during that study. For reference, the CalNex
campaign-average submicron $SAE$ for the 450-550 nm pair was 2.1. Thus, the conclusions here for the
CARES dataset are not necessarily applicable to the interpretation of the CalNex dataset.) Further, a fit
of the mean binned values of $E_{abs}$ extrapolated to $f_{ext,TD} = 0$ gives only $E_{abs} = 1.14$ ($\pm 0.02$) and 1.29 ($\pm 0.06$)
at 532 nm and 405 nm, respectively, suggesting that substantially larger values than the observed range
would not have been likely had a greater extent of evaporation been observed. The larger extrapolated
value at 405 nm than at 532 nm is consistent with a small contribution from so-called "brown carbon,"
which has an absorption spectrum that strongly increases towards shorter wavelengths, to the observed
absorption.
The particle single scatter albedo exhibits a non-monotonic dependence on $f_{ext,TD}$ at both 532 nm
and 405 nm (Figure 10e,f). On average, the ambient SSA values are at a minimum of 0.85 around $f_{ext,TD}$
= 0.35. The SSA then increases at either larger or smaller $f_{ext,TD}$. The increase in SSA towards smaller
$f_{ext,TD}$ likely reflects an increasing contribution of secondary aerosol species within the submicron mode
(e.g. organics, ammonium sulfate, ammonium nitrate) relative to BC, leading the overall particulates to
appear both more volatile and more scattering. Indeed, HR-AMS measurements indicate that secondary
inorganic and organic species are dominant components of the submicron particles in the Sacramento and
Sierra Nevada foothill region during CARES (Setyan et al., 2012; Shilling et al., 2013). The increase in



SSA towards larger $f_{ext,TD}$ likely results from the increasing contribution of non-volatile sea salt and dust
components within the supermicron mode that are either non- or very weakly absorbing. Looking at the
change in SSA upon heating in the TD, ΔSSA, there is a clear increase in ΔSSA with decreasing $f_{ext,TD}$
(Figure 10g,h). This is as expected because if material does not evaporate then no change in SSA should
be observed. The ΔSSA linearly extrapolated to $f_{ext,TD} = 0$ is 0.46 (±0.02) and 0.50 (±0.02) at 532 nm and
405 nm, respectively, corresponding to absolute extrapolated SSA values of ~0.4 given the observed
ambient particle SSA values. (Linear extrapolation to zero is not fully justifiable because the $f_{ext,TD}$ cannot
go to zero if there is some BC around and because there appears to be some flattening off in the ΔSSA
values at smaller $f_{ext,TD}$. Nonetheless, it can provide an estimate in the limit of small BC contributions.)
These extrapolated SSA values are relatively large compared to some laboratory observations for "fresh"
BC particles that are produced, for example, from flames (Cross et al., 2010) or gasoline or diesel vehicles
(Schnaiter et al., 2005; Forestieri et al., 2013) and that have little intrinsic organic material, but slightly
smaller than that reported by one other laboratory study on flame-generated soot (Radney et al., 2014).
Primary emitted BC has a fractal-like structure that is thought to collapse over time through atmospheric
ageing processes. This change in shape due to collapse is thought to lead to an increase in the SSA,
separate from any contributions from scattering coating materials (Chakrabarty et al., 2014). That the
extrapolated SSA values are larger than many of the laboratory studies on fresh BC suggests that the
sampled particles were somewhat collapsed compared to their emitted state.

For the period where the T0 site was impacted by particles emitted from local road surfacing

activities (e.g. asphalt), the ambient SSA values are small and the ΔSSA values deviate from the general
relationship observed for other periods. In fact, the absolute SSA measured for thermodenuded particles
during this particular period are around zero at 405 nm but ~0.2 at 532 nm. Such very small SSA values
suggest that absorption is dominated by very small particles, or at least particles that are agglomerates of
very small spherules; the surface area-weighted size distribution measured during this period peaked
around 300 nm diameter and the SP2 BC particle size distributions clearly indicate that the overall BC
particle size was larger during the asphalt-impacted period (Figure S9), suggesting that agglomerates of
small spherules is most plausible. We cannot entirely rule out the possibility that the measurements during
this period were strongly impacted by some absorbing gas-phase species (e.g. $NO_2$), confounding the SSA
measurements, although there was no evidence of gas-phase absorption in the background CRD channels
during this period suggesting that this is unlikely.



## *4 Conclusions*


Optical property measurements of $PM_1$, $PM_{2.5}$ and $PM_{10}$ made during the CARES 2010 field study
have been examined to develop understanding of the relationships between various intensive properties
and to establish differences in behavior between sub- and supermicron PM. Measurements were made at
two sites in the Sacramento region, one urban (T0) and one more rural (T1) but impacted by the urban
outflow on most days under southwesterly flow conditions (Fast et al., 2012). At both sites, there is a
strong contribution of supermicron particles to the total scattering, averaging around 50% at both sites.
The source of these supermicron particles appears to be a combination of local dust and sea spray, along
with some contributions from penetration of traditionally submicron particles into the supermicron mode.
The specific contributions of any of these supermicron particle sources varies with time and depends on
the prevailing transport patterns with, perhaps not surprisingly, generally larger sea spray contributions
when air masses have been transported from the San Francisco Bay Area. The measured scattering
Ångstrom exponents (*SAE*) for $PM_{10}$ are strongly correlated with the submicron versus supermicron
fraction of the total scattering, with similar linear relationships observed at both sites. This relationship
held despite there being variations in the size distributions within a given mode, which can theoretically
alter the *SAE*. This suggests that the relationships determined here are quite general, and that the *SAE* can
be used to quantitatively attribute scattering to sub- and supermicron particles. There was no notable
dependence of the absorption Ångstrom exponent (*AAE*) on *SAE* for $PM_{10}$, and these observations were
used to propose an updated particle classification scheme based on the relationship between these two
parameters.
The influence of photochemical processing on the sub- versus supermicron contribution to scattering
differed between the two sites, with photochemical processing leading to an increase in the submicron
fraction of scattering for the T0 (urban) site but minimal change, or even a slight decrease, at the T1
(downwind) site. This reflects in part the strong daytime peak in photochemical age at the T0 site in
contrast to the more gradual increase at the T1 site, coupled with the much stronger diurnal profile in the
wind speed, with a daytime peak, at the T1 site. Consequently, at the T1 site, photochemical production
of secondary PM was spread over a wider range of times due to transport and was countered through
local, temporally similar increases in dust production due to the higher daytime winds. At the T0 site, the
strong photochemical production of secondary PM led to a clear increase in the submicron fraction of
scattering with photochemical ageing.
The mass scattering coefficient for the supermicron particles varied inversely with the median
surface-weighted particle diameter ($d_{p,surf}$) of the supermicron mode, in general accordance with



theoretical expectations. This indicates clear temporal variability in the nature of the supermicron particle
sources, which seem to be coupled to the prevailing wind direction or air mass history, as established
through consideration of back trajectories. Light absorption was dominated by submicron particles,
although there was some contribution from the supermicron particles. The mass absorption coefficient for
supermicron particles exhibited a clear dependence on the supermicron $d_{p,surf}$, most likely due to variations
in the relative contributions of non-absorbing sea spray particles, penetration of BC from the submicron
mode, and very weak absorption by supermicron dust particles. Particle backscatter was found to be
related to the relative fractions of sub- versus supermicron scattering, but with an additional sensitivity to
variations in the size distribution within the submicron size range. The susceptibility of the particles to
heating in a thermodenuder depended explicitly on the contribution of supermicron particles to the $PM_{2.5}$
extinction, most likely because a large fraction of the supermicron particles were either essentially non-
volatile sea spray or dust particles. Heating in general led to an increase in the average particle
hygroscopicity and a decrease in the single scatter albedo. These together indicate that the residual
particles are likely a combination of absorbing submicron BC and somewhat hygroscopic supermicron
sea spray and less hygroscopic dust. The results presented here demonstrate that optical property
measurements can be used to assess likely chemical differences in the contributing particle types, and thus
to identify key PM sources.

## Author Information

Corresponding Author: Christopher D. Cappa
E-mail: cdcappa@ucdavis.edu
The authors declare no competing financial interest.

## Acknowledgements

This work was supported by the Atmospheric System Research (ASR) program sponsored by the
US Department of Energy (DOE), Office of Biological and Environmental Research (OBER), including
Grant No. DE-SC0008937. The authors acknowledge W. Berk Knighton for the PTR-MS data at the T1
site, R. Subramanian for the SP2 data, Ari Setyan for collection of the SMPS data at the T1 site and B.
Tom Jobson for the $NO_x$, $NO_y$, PTR-MS and meteorological data at the T0 site. The authors acknowledge
the NOAA Air Resources Laboratory (ARL) for the provision of the HYSPLIT transport and dispersion
model (http://www.ready.noaa.gov) used in this publication. The backscattering Mie calculations were
performed using MiePlot from Philip Laven (www.philiplaven.com/mieplot.htm). Funding for data





collection was provided by the US DOE's Atmospheric Radiation Measurement (ARM) Program. All
data used in this study are available from the ARM data archive at:
http://www.arm.gov/campaigns/aaf2009carbonaerosol. The views expressed in this document are solely
those of the authors and the funding agencies do not endorse any products or commercial services
mentioned in this publication.

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



**Table 1.** Table of instrumentation.

| Instrument | Property Measured | Site |
|---|---|---|
| UCD Photoacoustic Spectrometer (PAS)[a] | Dry $PM_{2.5}$ light absorption at 405 nm and 532 nm | T0 |
| UCD Cavity Ringdown Spectrometer (CRDS)[a] | Dry $PM_{2.5}$ light extinction at 405 nm and 532 nm; humidified particle extinction at 532 nm | T0 |
| Particle Soot Absorption Photometer (PSAP)[b] | Dry $PM_1$ and $PM_{10}$ light absorption at 470, 532 and 660 nm | T0, T1 |
| Nephelometer[b] | Dry $PM_1$ and $PM_{10}$ light scattering at 450, 550 and 700 nm | T0, T1 |
| Aerodyne High Resolution Time of Flight Aerosol Mass Spectrometer (HR-ToF-AMS) | Non-refractory $PM_1$ composition (NR-$PM_1$); Organic aerosol types through positive matrix factor analysis | T0, T1 |
| PNNL Single Particle Laser Ablation Time of Flight Mass Spectrometer (SPLAT-II) | Single particle composition and identification for $PM_{0.05}$-$PM_2$ (optimized for $PM_{0.1-0.6}$) | T0 |
| Particle Ablation Laser-desorption Mass Spectrometer (PALMS) | Single particle composition and identification for $PM_{0.15}$-$PM_2$ | T1 |
| Single Particle Soot Photometer (SP2) | Refractory black carbon (rBC) number and mass concentration and size distributions | T0, T1 |
| Scanning mobility particle sizer (SMPS) | $PM_1$ particle mobility size distributions | T0, T1 |
| Aerodynamic particle sizer (APS) | $PM_{0.7}$-$PM_{10}$ aerodynamic size distributions | T0, T1 |
| $NO_x$ chemiluminescence | $NO + NO_2$ (gas-phase) | T0 |
| $NO_y$ by thermal conversion and chemiluminescence | Nitrogen oxides ($NO + NO_2 + HNO_3$ + alkyl nitrates + peroxy nitrates) | T0 |
| Proton Transfer Reaction Mass Spectrometer (PTR-MS) | Concentrations of select volatile organic compounds (specifically, benzene and toluene) | T0, T1 |

[a]These instruments sampled either ambient particles or particles that had been thermodenuded at 225 °C, switching on a 2.5 or 5 minute cycle
[b] These instruments alternately sampled $PM_1$ or $PM_{10}$ on a 6 minute cycle







**Table 2.** Campaign average optical properties at T0 and T1 for submicron and supermicron particles.


| Property | T0 | | | T1 | |
| --- | --- | --- | --- | --- | --- |
| | $PM_1$ | $PM_{2.5}$ | supermicron[+] | $PM_1$ | supermicron[+] |
| $b_{\text{sca},450}$ | 12.9 Mm$^{-1}$ | | 10.7 Mm$^{-1}$ | 12.7 Mm$^{-1}$ | 6.6 Mm$^{-1}$ |
| $b_{\text{sca},550}$ | 7.9 Mm$^{-1}$ | | 10.9 Mm$^{-1}$ | 7.6 Mm$^{-1}$ | 6.8 Mm$^{-1}$ |
| $b_{\text{sca},700}$ | 4.2 Mm$^{-1}$ | | 10.9 Mm$^{-1}$ | 3.4 Mm$^{-1}$ | 6.8 Mm$^{-1}$ |
| $b_{\text{abs},470}$ | 2.3 Mm$^{-1}$ | | 0.31 Mm$^{-1}$ | 1.45 Mm$^{-1}$ | 0.26 Mm$^{-1}$ |
| $b_{\text{abs},530}$ | 1.9 Mm$^{-1}$ | | 0.25 Mm$^{-1}$ | 1.25 Mm$^{-1}$ | 0.20 Mm$^{-1}$ |
| $b_{\text{abs},660}$ | 1.5 Mm$^{-1}$ | | 0.19 Mm$^{-1}$ | 0.95 Mm$^{-1}$ | 0.14 Mm$^{-1}$ |
| $b_{\text{ext},405}$ | | 27.0 Mm$^{-1}$ | | | |
| $b_{\text{ext},532}$ | | 18.0 Mm$^{-1}$ | | | |
| $b_{\text{abs},405}$ | | 2.8 Mm$^{-1}$ | | | |
| $b_{\text{abs},532}$ | | 2.1 Mm$^{-1}$ | | | |
| $AAE_{470\text{-}532}$[#] | 1.21 ± 0.18 | | 1.93 ± 0.83 | 1.22 ± 0.33 | 2.03 ± 1.04 |
| $AAE_{470\text{-}660}$[#] | 1.17 ± 0.11 | | 1.54 ± 0.50 | 1.28 ± 0.22 | 1.76 ± 0.69 |
| $AAE_{532\text{-}660}$[#] | 1.15 ± 0.12 | | 1.30 ± 0.52 | 1.28 ± 0.21 | 1.68 ± 0.77 |
| $AAE_{405\text{-}532}$[#] | | 1.3 ± 0.9 | | | |
| $SAE_{450\text{-}550}$[*] | 2.42 ± 0.38 | | -0.13 ± 0.31 | 2.58 ± 0.27 | -0.15 ± 0.34 |
| $EAE_{405\text{-}532}$[*] | | 1.53 ± 0.5 | | | |

[+] Values for supermicron particles are calculated as the difference between $PM_{10}$ and $PM_1$.
[#] The reported uncertainties were determined from fitting a histogram of the observed values to a Gaussian distribution and are the 1σ spread.
[*] The reported uncertainties are 1σ standard deviations.









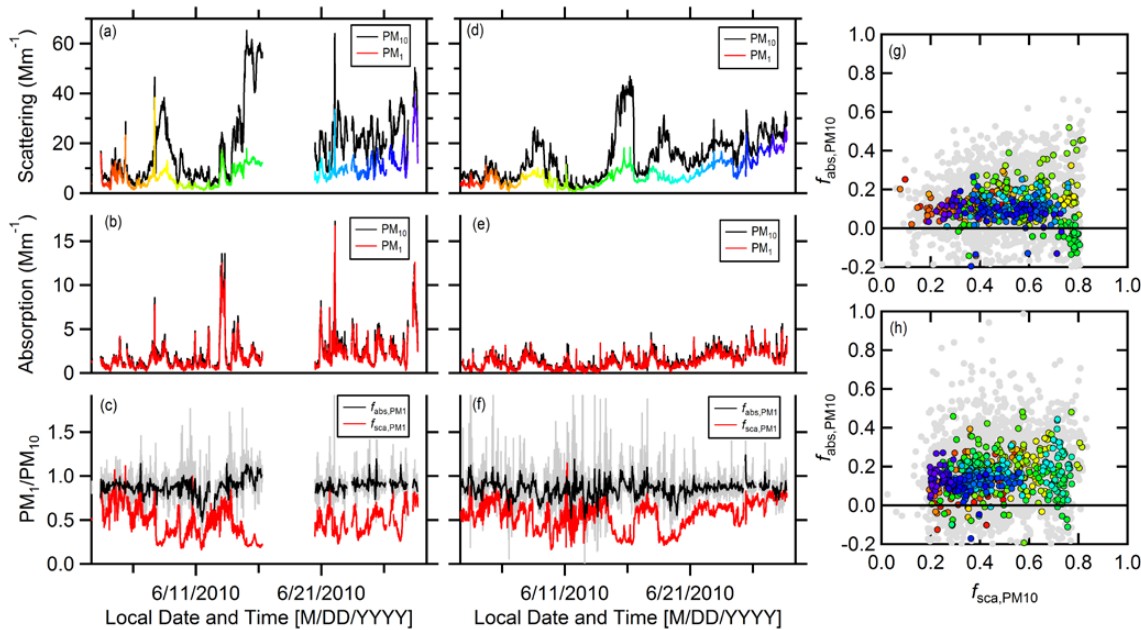


**Figure 1.** Time-series of $PM_1$ and $PM_{10}$ scattering (a and d) and absorption (b and e) coefficients at 550 nm for T0 (left panels) and T1 (right panels). Values for $PM_{10}$ are shown as black lines and for $PM_1$ as colored lines. The ratio between $PM_1$ and $PM_{10}$ scattering (red) and absorption (black) are shown in panels c and f. (For absorption, the data have been further averaged to 1 hour; the higher time resolution data are shown as gray.) The co-variation between $f_{abs,PM10}$ and $f_{sca,PM10}$ for T0 (g) and T1 (h) are also shown. The 1-hr averaged points are colored according to time during the campaign, and correspond to the time-series in panels a and d; the gray points are the data at higher time-resolution.

1024




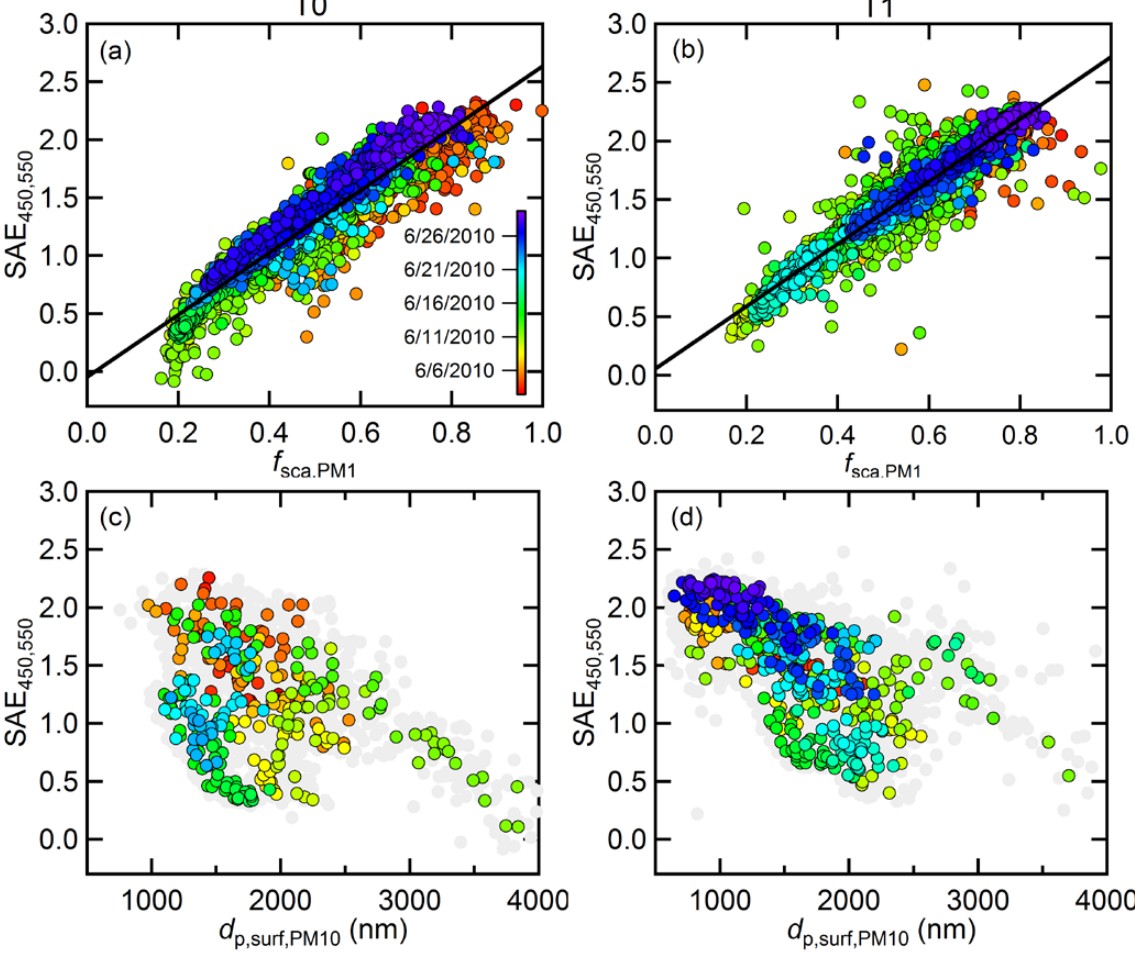

1025

**Figure 2.** (a,b) The relationship between the scattering Ångstrom exponent for the 450-550 nm pairs and the $f_{\text{sca,PM1}}$ for both T0 (left panels) and T1 (right panels). (c,d) The relationship between the SAE and the median surface-weighted diameter for $PM_{10}$. The points in all graphs are colored according to time during the campaign (see legend). For panels (a,b) data at 10 min resolution are shown, while in panels (c,d) the colored points are for data averaged to 1 h and the gray points are for 10 min data. The fewer colored points in panel (c) is the result of a malfunction of the APS after 22 June 2010, which precludes calculation of $d_{\text{p,surf,PM10}}$.

1033

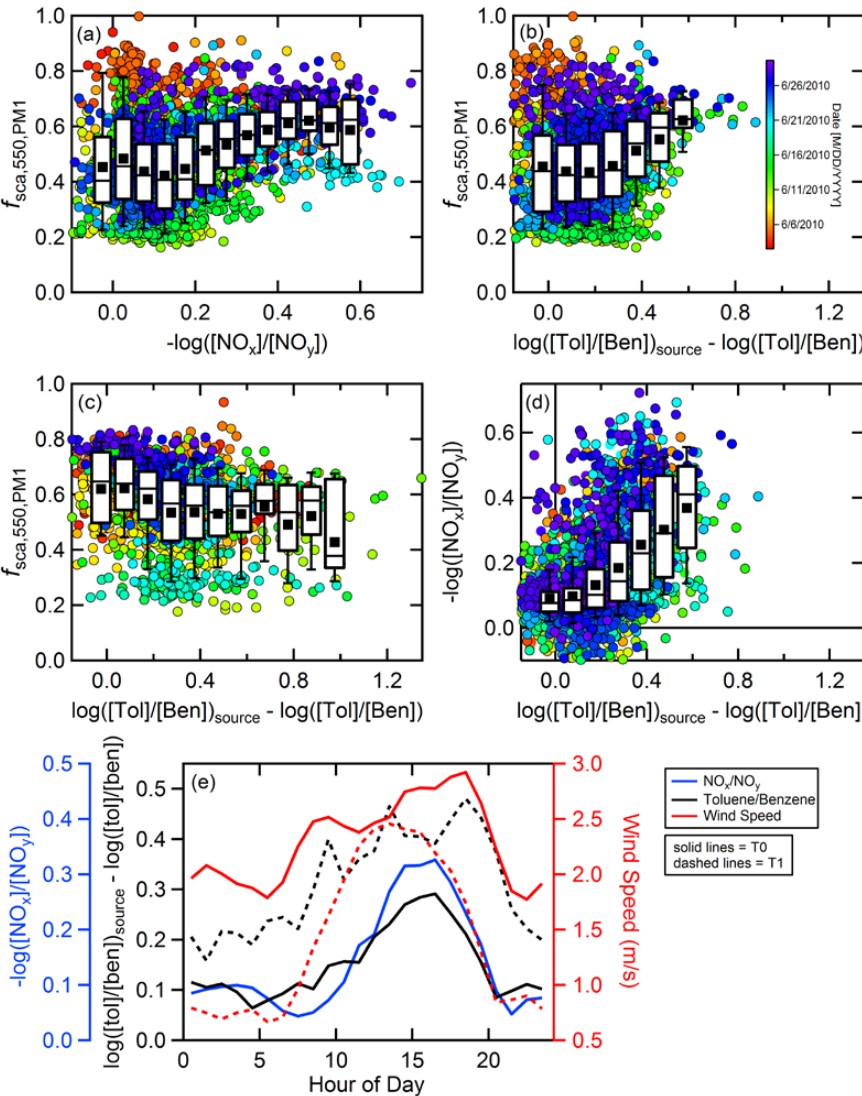

**Figure 3.** Submicron fraction of scattering for the T0 and T1 sites as a function of photochemical age proxies. Observations at T0 using the (a) $NO_x/NO_y$ ratio and the (b) toluene/benzene ratio and at T1 for the (c) toluene/benzene ratio as the PCA proxy. Individual measurements (averaged to 10 minutes) are colored by time. (d) The relationship between the two PCA estimation methods at T0. (e) The diurnal variation in the PCA estimation methods and the measured wind speed for T0 (solid lines) and T1 (dashed lines). Box and whisker plots show the median (line), mean (square) upper and lower quartile (box) and $10^{th}$ and $90^{th}$ percentile (whiskers).




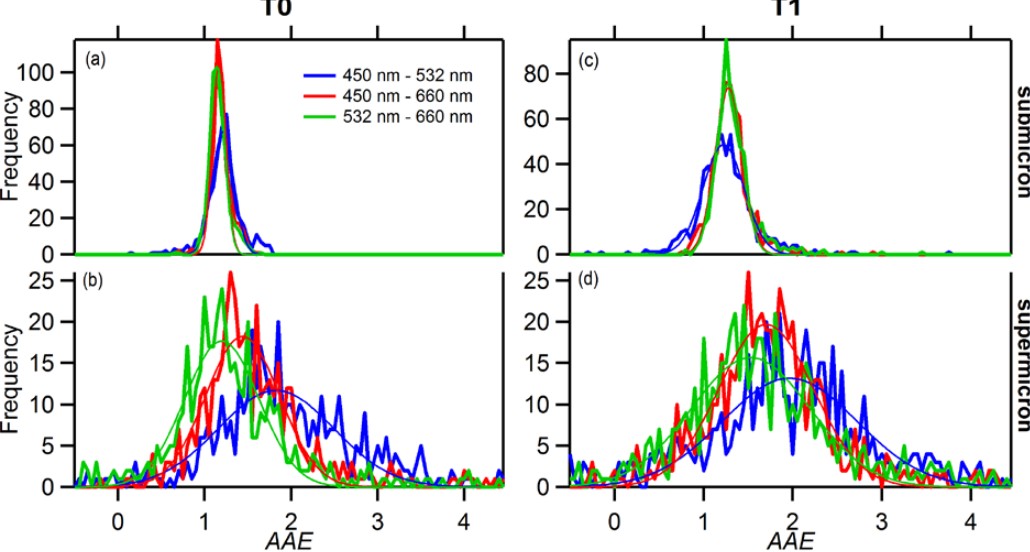



**Figure 4.** Histograms of the measured *AAE* values for various wavelength pairs for submicron (top; a and
c) and supermicron (bottom; b and d) particles at the T0 (left) and T1 (right) sites. The different colors
correspond to different wavelength pairs (see legend). The thick lines correspond to the observations while
the thin lines show the results from fitting of the distributions to a Gaussian function.





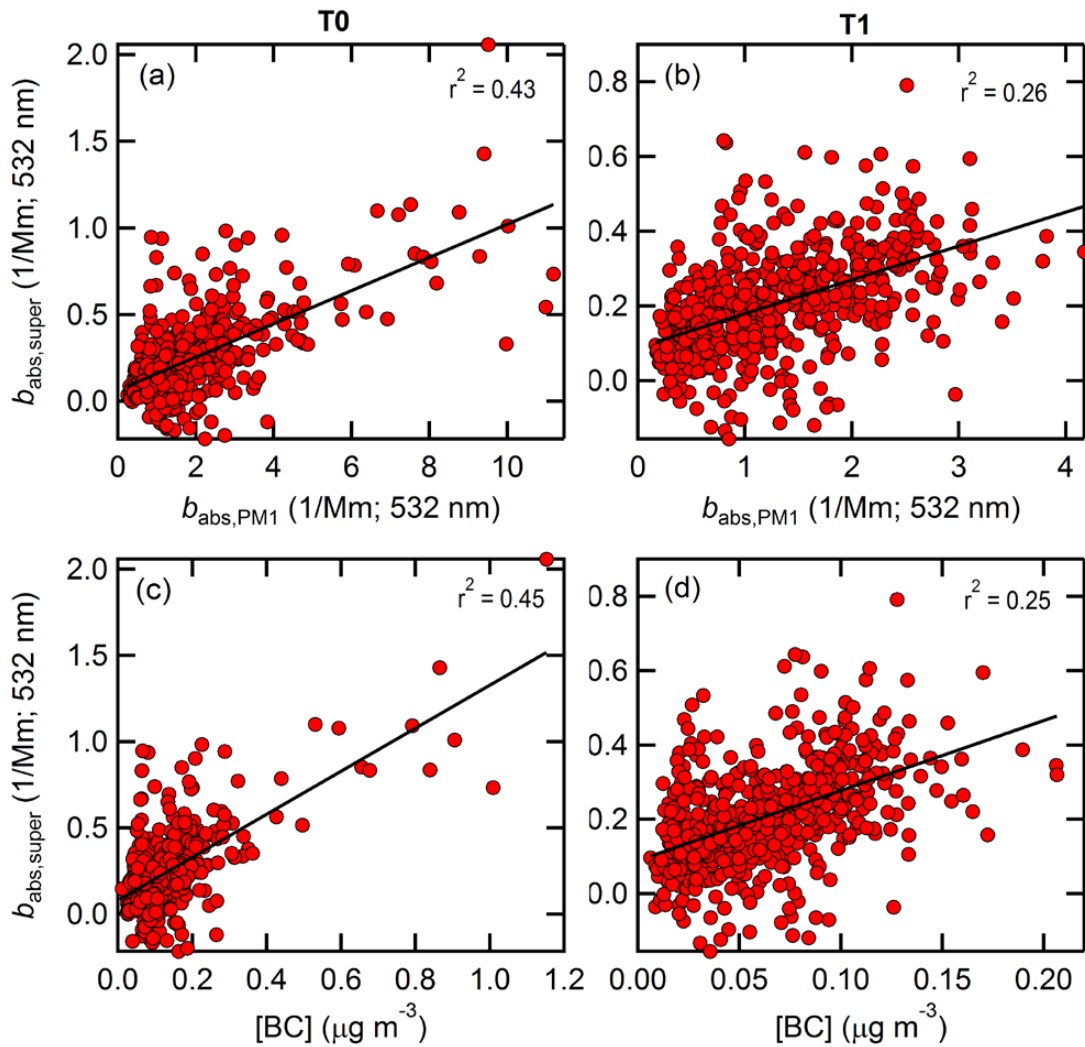



**Figure 5.** Co-variation of the supermicron absorption at T0 (left panels) and T1 (right panels) with the submicron absorption (top panels) and with black carbon concentration (bottom panels). Note that negative values of $b_{abs,super}$ result from this being derived from the difference between $b_{abs,PM10}$ and $b_{abs,PM1}$.












**Figure 6.** (a,b) Scatter plot between $b_{\text{sca,super}}$ at 532 nm and [PM$_{\text{super}}$] for (left panels) T0 and (right panels)
T1. The lines correspond to different *MSC* values (in m$^2$ g$^{-1}$). (c,d) Scatter plot between $b_{\text{abs,super}}$ at 532 nm
and [PM$_{\text{super}}$]. The lines correspond to different *MAC* values (in m$^2$ g$^{-1}$). (e,f) The relationship between 1-
hr average *MSC* values and the surface area weighted mean diameter, $d_{\text{p,surf}}$ at the two sites. (g,h) The
relationship between 1-hr average *MAC* values and $d_{\text{p,surf}}$ at the two sites. The individual 1-hr average data
points are shown overlaid by box-and-whisker plots showing the mean (■), median (-), lower and upper
quartile (boxes), and 9$^{\text{th}}$ and 91$^{\text{st}}$ percentile (whiskers). The points in panels a-h are colored according to
time, and correspond to the colors in the bottom figure and color scale. (i) Time series of the supermicron
particle mass concentration for T0 and T1. T0 values are black lines, and T1 values are colored according
to time.






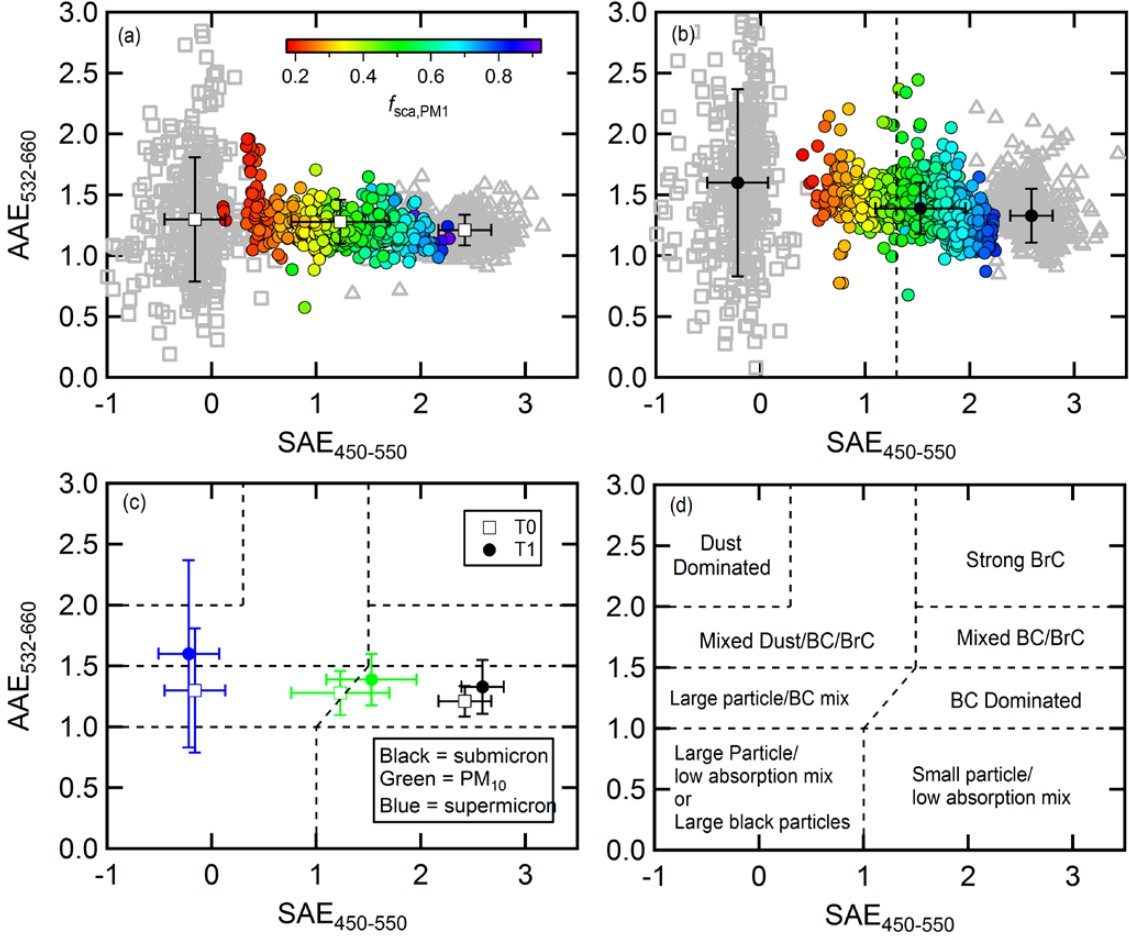


**Figure 7.** Observed relationship between the *AAE* (532-600 nm pair) and the *SAE* (450-550 nm pair) for
PM$_{10}$ (colored circles), submicron (open grey triangles) and supermicron (open grey squares) particles
for the (a) T0 and (b) T1 sites. (c) Comparison between the PM$_{10}$ (green), submicron (black) and
supermicron (blue) particle averages between the T0 (filled circles) and T1 (open squares) sites. (d) An
alternate classification scheme to that suggested by Cazorla et al. (2013).

1077



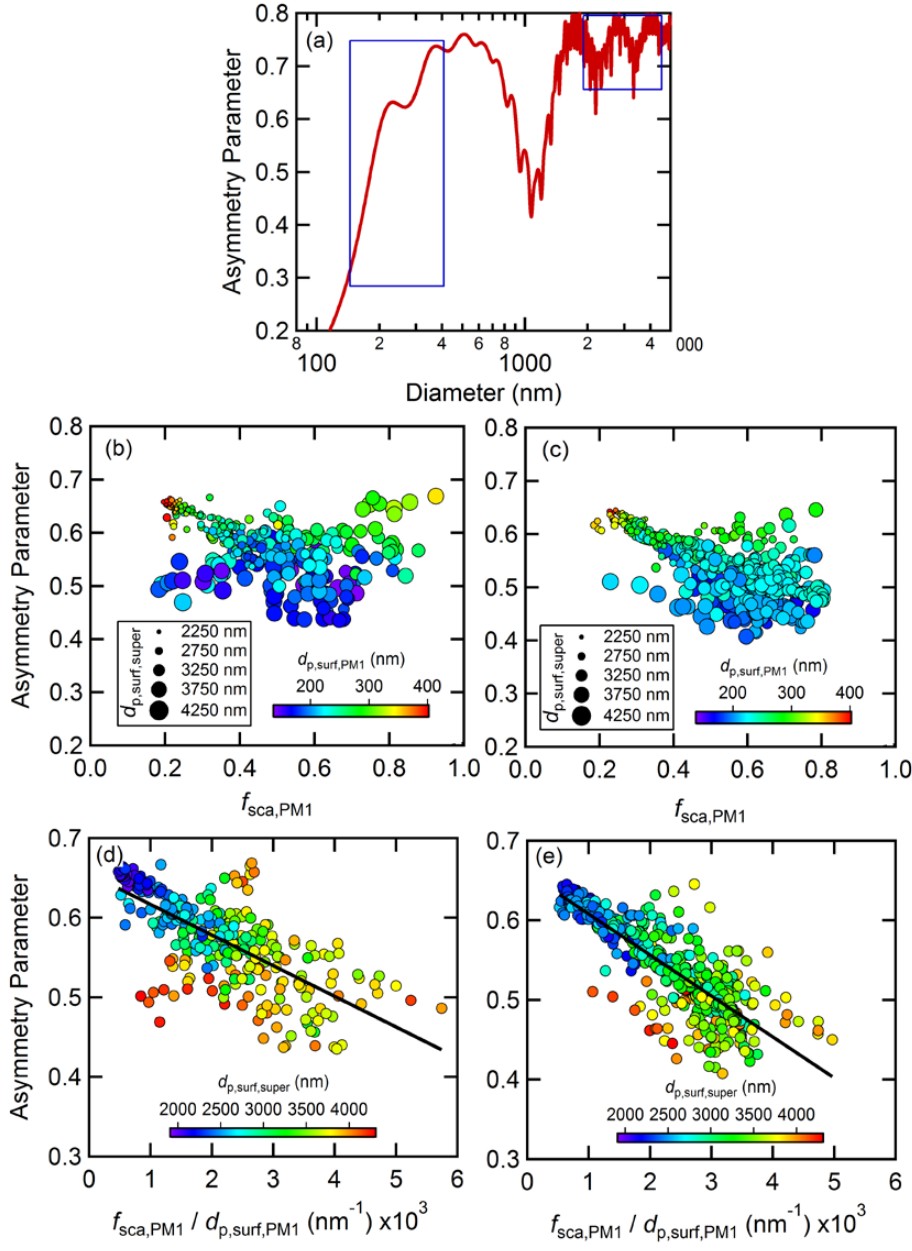

1078

**Figure 8.** (a) Theoretical variation in the asymmetry parameter, $g_{sca}$, with particle diameter, assuming

spherical particles with RI = 1.5 + 0.0$i$. The blue boxes indicate the range of $d_{p,surf}$ values observed for

submicron and supermicron particles. (b,c) The observed dependence of $g_{sca}$ on $f_{sca,PM1}$ for (b) T0 and (c)

T1. The points are colored according to $d_{p,surf,PM1}$ and the point size corresponds to $d_{p,surf,super}$. (d,e) The

relationship between $g_{sca}$ and $R_g = f_{sca,PM1}/d_{p,surf,PM1}$ for (d) T0 and (e) T1. The points are colored

according to $d_{p,surf,super}$.




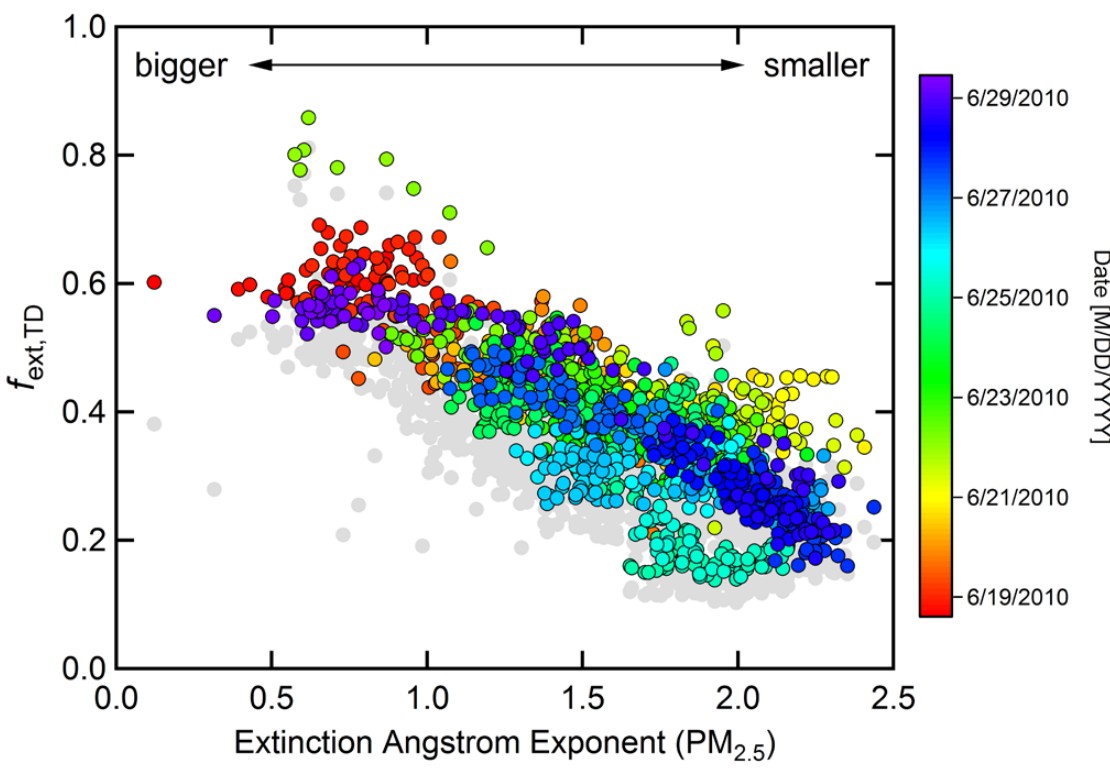


**Figure 9.** Variation of the extinction fraction remaining at 532 nm (colored points) and 405 nm (gray points) as a function of the observed ambient particle extinction Ångstrom exponent for PM$_{2.5}$. The points are colored by date.







**Figure 10.** Dependence of various intensive parameters on the fraction of extinction remaining after heating in the thermodenuder. Data are shown for (a) the absolute $\gamma_{RH}$, (b) the change in $\gamma_{RH}$ (c) the absorption enhancement at 532 nm, (d) the absorption enhancement at 405 nm, (e) the SSA at 532 nm, (f) the SSA at 405 nm, (g) the change in SSA at 532 nm and (h) the change in SSA at 405 nm. The points are colored by time. The circled points in (a) show the period that was impacted by local road resurfacing activities.