# Peer review of "Understanding the Optical Properties of Ambient"

_Atmospheric Chemistry and Physics, 2015_

## Referee Comment (RC1) · Anonymous Referee #2 · 5 Feb 2016

This paper describes measurements of absorption, scattering, and extinction for PM1, PM2.5, and PM10 particles at two field sites near Sacramento, California. The authors use these measurements to conclude that supermicron particles contribute approximately half of scattering, and are composed of varying amounts of dust and sea salt. Photochemical processing does not have a consistent effect on submicron aerosol scattering, partly due to transport. The authors propose relationships between other intensive aerosol properties.

This is a well-written paper, although the discussion is long and could possibly ben-

efit from some condensing. I recommend publication after the following issues are addressed.

Major comments:

1. The introduction is short and does not summarize existing knowledge about the composition and optical properties of supermicron aerosol. A previous study from the CARES campaign has already reported the unexpectedly large contribution of coarse mode aerosol to radiative forcing (Kassianov et al., 2012). This paper and other relevant results (possibly including Malm et al., 1994; Dubovik et al., 2002; Hand et al., 2002; Malm et al., 2007; Eck et al., 2010) should be described and cited in an expanded introduction.

2. Lines 156-157: "Data during the first week of the campaign (June 3-12) are especially noisy due to instrumental problems." What caused the noise and were the measurements still accurate?

3. Section 2.2: What were the specific differences between the T0 and T1 site? Were the HR-AMS instruments operated by the same research group? Do the different OOA mass factors represent true aerosol composition differences between the sites?

4. Section 3.1: Besides Kassianov e tal. (2012), what have prior studies of supermicron aerosol extinction under relatively clean conditions observed?

Minor comments:

Line 129: Give model and manufacturer for SMPS.

Line 176: What is the part number of the NOx chemiluminescence instrument?

Typographic corrections:

Line 32: "... but the there is some"

Line 79: Use SI units.
Lines 99,125, 203, 205: Remove comma after June

Line 157: This is a very long week.

References:

Dubovik, O., Holben, B., Eck, T. F., Smirnov, A., Kaufman, Y. J., King, M. D., Tanre, D., and Slutsker, I.: Variability of absorption and optical properties of key aerosol types observed in worldwide locations, J Atmos Sci, 59, 590-608, 2002.

Eck, T. F., Holben, B. N., Sinyuk, A., Pinker, R. T., Goloub, P., Chen, H., Chatenet, B., Li, Z., Singh, R. P., Tripathi, S. N., Reid, J. S., Giles, D. M., Dubovik, O., O'Neill, N. T., Smirnov, A., Wang, P., and Xia, X.: Climatological aspects of the optical properties of fine/coarse mode aerosol mixtures, J Geophys Res-Atmos, 115, 2010.

Hand, J. L., Kreldenweis, S. M., Sherman, D. E., Collett, J. L., Hering, S. V., Day, D. E., and Malm, W. C.: Aerosol size distributions and visibility estimates during the Big Bend regional aerosol and visibility observational (BRAVO) study, Atmos Environ, 36, 5043-5055, 2002. Kassianov, E., Pekour, M., and Barnard, J.: Aerosols in central California: Unexpectedly large contribution of coarse mode to aerosol radiative forcing (vol 39, L20806, 2012), Geophys Res Lett, 39, 2012.

Malm, W. C., Sisler, J. F., Huffman, D., Eldred, R. A., and Cahill, T. A.: Spatial and Seasonal Trends in Particle Concentration and Optical Extinction in the United-States, J Geophys Res-Atmos, 99, 1347-1370, 1994.

Malm, W. C., Pitchford, M. L., McDade, C., and Ashbaugh, L. L.: Coarse particle speciation at selected locations in the rural continental United States, Atmos Environ, 41, 2225-2239, 2007.

---

## Referee Comment (RC2) · Anonymous Referee #1 · 9 Feb 2016

This paper comprehensively reports the results of optical properties measurements made as part of the CARES campaign. This is a very weighty paper and while the conclusions could really be regarded as game-changing, the high quality of the measurements and the depth of the analysis means that these results are still very important and relevant to ACP, having a wide range of potential applications in radiative transfer and the interpretation of remote sensing data. I would recommend that this be published subject to minor corrections.

General: I found the various combinations of sampling conditions referred to a bit bewildering at times. It would greatly help the reader if a schematic figure could be given showing the different sampling arrangements for the different sites. On a related note, I also found the large number of mathematical symbols to be a bit confusing at times, so it would be useful to compile these as a table.

Line 122: The method used to humidify the sample flow to 85%, including the methods to monitor and control the humidity, should be described here.

Line 128: A reference should be supplied for the mobility conversion for the aerodynamic diameters. Rather than assume sphericity (which is probably not valid for dust and dry sea salt particles), it would probably be more correct to refer to the assumed 2 g/cc density as the 'effective' density.

Line 138: These mobility diameters should be qualified as 'approximate' because they have each been clearly rounded off compared to the actual theoretical values.

Line 166: The factor of 1.66 is contrary to the factor of 1.33 recommended by Laborde et al. (2012b), so the nature of the 'personal communication' cited may need expanding on here.

Line 233: A reference should be included here because the exact effect BrC has on AAE is by no means certain.

Line 238: Similarly, include references for examples of how SAE is 'commonly used'.

Line 384: What AAE would be necessary to cancel out any influence from BrC? How does this compare with DOI:10.1002/2014GL062443?

Line 402: A more fundamental reason the PALMS and SPLAT II are not capable of quantifying contributions from BC internally mixed on dust particles is the matrix effects associated with each instrument, such that the mass fractions reported on individual particles are not quantitative.

Line 453: Shouldn't the qualitative influence of sea salt particles be evident in the single

particle mass spectrometer data?

Line 514 (and elsewhere): "consistent with that of (Russell et al., 2010)" should read "consistent with that of Russell et al. (2010)"

Line 531: This opening statement should be made more descriptive, as it isn't specified what exactly the effect on climate is. It may also be worth mentioning that this is an important parameter for remote sensing retrievals.

Line 573: Was any correction for thermophoretic losses invoked? If not, the authors should comment on how much of an issue they believe this to be.

Line 593: According to conventional wisdom, supermicron sulphate and nitrate tends to be in the form of salts of calcium and sodium rather than ammonium, which makes them quite involatile, so I would doubt that this is significant.

Line 599: A related hypothesis could be that undenuded sea salt particles do not completely effloresce during drying due to the presence of magnesium salts (doi: 10.5194/acp-15-11273-2015) and organics. If the denuded particles are more completely dried out (due to the water being boiled off) then this would increase their apparent hygroscopicity further.

Line 620: How does this fit in with the conclusions of Doi:10.1038/Ngeo2220?

Line 646: The caveat should be added that this is assuming that the particles have not acquired an involatile coating (e.g. coagulation with sea salt, condensation of humic-like SOA), because this could confound any attempt to isolate the effect of morphological changes on SSA.

Line 653: It should be noted that if the particles are thought to be very non-spherical, the SMPS sizing is likely to be overestimated to a large degree

Line 653: If the particles are fundamentally different to 'normal' black carbon particles, then the SP2's calibration could be invalid and the equivalent core sizes reported

inaccurate. If the instrument had a narrow band incandescence detector, it may be informative to compare the ratio of this to the broadband detector to see if the apparent colour temperature had changed.

Figure S1: Adjust the colour scale so that areas covered by land are green rather than blue.

Figure S9: Given that Babs of black carbon is more closely related to mass than number, a mass-weighted distribution comparison would be informative here.

---

## Referee Comment (RC3) · Anonymous Referee #1 · 9 Feb 2016

Just noticed that there was a typo in my opening remarks. It should be "the conclusions could not really be regarded as game-changing". Apologies for letting this slip.

---

## Author Comment (AC2) · 30 Apr 2016

**Response to Comments from Anonymous Referee #2**

Response to comments on "Understanding the Optical Properties of Ambient Sub- and Supermicron Particulate Matter: Results from the CARES 2010 Field Study in Northern California" by C. D. Cappa et al.

We thank the reviewer for her/his comments, which have helped us to improve our work. The original reviewer comments are in **black** and our responses are in **blue**.

This paper describes measurements of absorption, scattering, and extinction for PM1, PM2.5, and PM10 particles at two field sites near Sacramento, California. The authors use these measurements to conclude that supermicron particles contribute approximately half of scattering, and are composed of varying amounts of dust and sea salt. Photochemical processing does not have a consistent effect on submicron aerosol scattering, partly due to transport. The authors propose relationships between other intensive aerosol properties. This is a well-written paper, although the discussion is long and could possibly benefit from some condensing. I recommend publication after the following issues are addressed.

*Major comments:*

1. The introduction is short and does not summarize existing knowledge about the composition and optical properties of supermicron aerosol. A previous study from the CARES campaign has already reported the unexpectedly large contribution of coarse mode aerosol to radiative forcing (Kassianov et al., 2012). This paper and other relevant results (possibly including Malm et al., 1994; Dubovik et al., 2002; Hand et al., 2002; Malm et al., 2007; Eck et al., 2010) should be described and cited in an expanded introduction.

We have expanded the introduction to some extent. We now note that while measurements of $PM_1$ and $PM_{10}$ composition and mass (which allows determination of supermicron particle properties) have become routine, the measurements of supermicron particle optical properties (via difference between $PM_{10}$ and $PM_1$ measurements) remain relatively uncommon. This goes directly to studies such as Kassianov et al. (2012), in which optical properties were inferred from size distribution measurements. Our paper reports on actual measurement of the supermicron optical properties and does not rely on "reconstruction" of optical properties from mass or composition measurements. We have added the following to the introduction:

"Measurements of particulate mass concentrations and composition for $PM_{2.5}$ and $PM_{10}$ have become routine through networks such as the U.S. Interagency Monitoring of Protected Visual Environments (IMPROVE) network (Malm et al., 2007), and these can be used to "reconstruct" aerosol optical properties (Malm and Hand, 2007). However, direct measurements of the optical properties of particles between different size regimes are much less common, and where they do exist are quite often made in the marine boundary layer (e.g. Bates et al., 2006) and not over land. ($PM_{2.5}$ and $PM_{10}$ refer to particulate matter with aerodynamic diameters below 2.5 μm and 10 μm, respectively.)"

"A previous analysis of particle size distributions measured during the CARES campaign indicated a large contribution of supermicron aerosol to the total particle scattering (Kassianov et al., 2012). Here, direct measurements of the scattering by these supermicron particles are reported on, and their sources and properties and the factors that drive their variability are examined."

"Results from *in situ* measurements such as here can help to inform remote sensing retrievals and climatologies, which can provide a much broader spatial picture of sub- versus supermicron abundances and contributions to light scattering and extinction (Dubovik et al., 2002; Eck et al., 2010)."

2. Lines 156-157: "Data during the first week of the campaign (June 3-12) are especially noisy due to instrumental problems." What caused the noise and were the measurements still accurate?

The noisy AMS data during the first week occurred because the AMS was inadvertently set wrong during the first week, with a very short MS-open duration and long MS-closed duration. This uneven open/closed cycle was the cause of noisier data (decreased precision) but did not affect quantification. The AMS was returned to normal setting of equal durations for MS open and MS closed mode June 16 onwards.

3. Section 2.2: What were the specific differences between the T0 and T1 site? Were the HR-AMS instruments operated by the same research group? Do the different OOA mass factors represent true aerosol composition differences between the sites?

There are many differences in location, available instrumentation and operators between the T0 and T1 sites. All of these differences are detailed in the "Overview of the 2010 Carbonaceous Aerosols and Radiative Effects Study (CARES)" by Zaveri et al. (2012), which is referenced in the overview of the Experimental section. As specified in Zaveri et al. (2012), the HR-ToF-AMS was operated by Qi Zhang's group from UC Davis at the T1 site and by Chen Song and Rahul Zaveri's group from Pacific Northwest National Laboratory at the T0 site. Regardless, OOA never represents "true" aerosol composition differences, only a mathematical representation of those differences that has some physical/chemical interpretation. However, the reviewer is most likely asking about comparability between OOA from the two sites. The HR-AMS measurements from T0 and T1 are discussed at length in Setyan et al. (2014), and thus rather than repeating those results here, we refer the reader to that work for more details. But the short answer is "yes, the differences in OOA factors are meaningful."

4. Section 3.1: Besides Kassianov et al. (2012), what have prior studies of supermicron aerosol extinction under relatively clean conditions observed?

To be clear, Kassianov et al. (2012) calculated supermicron aerosol extinction from size distributions, while in the current study we actually report on measurements of the scattering, absorption and extinction by the supermicron particles in this region. As we now note in the introduction, direct measurements of supermicron optical properties are fairly uncommon, and where they do exist are most often found for the marine boundary layer where the contribution of sea spray will be large. We have focused our revisions on studies in which optical properties

were measured (either *in situ* or remote sensing) and not considered measurements of particulate mass concentrations alone. To this end, we have added an extensive comparison with some remote sensing results that look at the relationship between Angstrom exponents and the fine mode fraction of extinction, and that look at the relationship between the SSA and the fine mode fraction of extinction. We have added two new figures in support, one in the supplemental and one in the main text.

The added text and figures are provided below.

[revised manuscript text omitted]

*Minor comments:*

Line 129: Give model and manufacturer for SMPS.

Done.

Line 176: What is the part number of the NOx chemiluminescence instrument?

We now give further information about the operation of the instrument, specifically stating "Gas-phase concentrations of the sum of NO and $NO_2$ (= $NO_x$) and the sum of nitrogen oxides (= $NO_y$) were measured using a 2-channel chemiluminescence instrument (Air Quality Design, Inc, High Performance, 2-Channel) in which $NO_2$ is photolyzed to NO *using a blue light photolytic converter* and $NO_y$ is converted to NO on a Mo catalyst," and where the new text is in italics.

*Typographic corrections:*

Line 32: "... but the there is some" - done

Line 79: Use SI units. - done

Lines 99,125, 203, 205: Remove comma after June - done

Line 157: This is a very long week. – We now indicate this is a week and a half

---

## Author Response (AR1)

Dear Dr. Cziczo,

Below you will find our point-by-point responses to each of the reviewers, including an indication of changes made in response to their comments. In addition, you will find an annotated "track changes" version of the manuscript. We have made numerous changes to the manuscript in response to the reviewer comments, and believe that we have addressed their concerns in full. We thank you for your consideration of this work.

Regards,

Chris Cappa

UC Davis

**Response to Comments from Anonymous Referee #1**

Response to comments on "Understanding the Optical Properties of Ambient Sub- and Supermicron Particulate Matter: Results from the CARES 2010 Field Study in Northern California" by C. D. Cappa et al.

We thank the reviewer for her/his comments, which have helped us to improve our work. The original reviewer comments are in **black** and our responses are in **blue**.

This paper comprehensively reports the results of optical properties measurements made as part of the CARES campaign. This is a very weighty paper and while the conclusions could not really be regarded as game-changing, the high quality of the measurements and the depth of the analysis means that these results are still very important and relevant to ACP, having a wide range of potential applications in radiative transfer and the interpretation of remote sensing data. I would recommend that this be published subject to minor corrections.

General: I found the various combinations of sampling conditions referred to a bit bewildering at times. It would greatly help the reader if a schematic figure could be given showing the different sampling arrangements for the different sites. On a related note, I also found the large number of mathematical symbols to be a bit confusing at times, so it would be useful to compile these as a table.

A table has been added to aid the reader in recalling the definition of symbols used throughout the manuscript (new Table S1). A schematic has also been added to the supplemental material (new Figure S2).

Line 122: The method used to humidify the sample flow to 85%, including the methods to monitor and control the humidity, should be described here.

The following sentence has been added to address the method used for humidification:

"The air stream was humidified by passing it through a custom humidifier, which consisted of a semi-permeable capillary membrane (Accurel) that was kept continuously wetted. The relative humidity in the CRD cells was monitored using Vaisala RH probes (HMP50) that were calibrated using saturated salt solutions."

Line 128: A reference should be supplied for the mobility conversion for the aerodynamic diameters. Rather than assume sphericity (which is probably not valid for dust and dry sea salt particles), it would probably be more correct to refer to the assumed 2 g/cc density as the 'effective' density.

A reference has been added (DeCarlo et al., 2004) and we now use the "effective density" terminology.

Line 138: These mobility diameters should be qualified as 'approximate' because they have each been clearly rounded off compared to the actual theoretical values.

The sentence now reads:

"The mobility equivalent cut-diameters are (assuming a density of 2 g cm$^{-3}$) *approximately* 700 nm, 1750 nm and 7200 nm, respectively."

Line 166: The factor of 1.66 is contrary to the factor of 1.33 recommended by Laborde et al. (2012b), so the nature of the 'personal communication' cited may need expanding on here.

First, we note that we used 1.53, not 1.66. Second, we note that in Laborde et al., the relationship between Aquadag and Fullerene soot is not 1.33, but a larger factor. From their Fig. 5, using a BC particle mass of 10 fg as a reference point, the scaling between these two is about 1.35/0.9 = 1.5, which is very similar to the value of 1.53 used here. Finally, we have added the sentence "This adjustment factor was determined from laboratory studies conducted after the CARES campaign."

Line 233: A reference should be included here because the exact effect BrC has on AAE is by no means certain.

We fully agree that the effect of BrC on AAE values is "by no means certain." At the same time, it seems quite clear that BrC tends to increase the AAE over pure BC. We have added a reference to Lack and Cappa (2010) here, although there are many references that could be chosen to illustrate this point.

Line 238: Similarly, include references for examples of how SAE is 'commonly used'.

We have added a reference to (Clarke and Kapustin, 2010), which summarizes results from 14 different flight campaigns using SAE as a key criteria for particle size. Many references could be chosen, so we have taken the approach of selecting just one that considers a multitude of different data sets.

Line 384: What AAE would be necessary to cancel out any influence from BrC? How does this compare with DOI:10.1002/2014GL062443?

For BrC to be entirely cancelled out, the AAE for BC would need to simply be assumed equal to the observed AAE, namely 1.17 at T0 and 1.28 at T1 (from Table 2). We have modified the sentence to read: "if the actual $AAE_{BC}$ were >1, as possibly suggested by the $AAE_{532-660}$ measurements at both sites, then the attributed brown carbon fraction would be even smaller *and cancelled out entirely if AAE$_{BC}$ equals the observed value for the ambient particles.*"

Regarding the comparison with Liu et al. (2015), we are not entirely certain what the reviewer is asking us to compare. The Liu et al. (2015) study is primarily a theoretical study when it comes to consideration of the AAE. What we are discussing is really an empirical approach to assessing BrC properties. In other words, an AAE is assumed for BC, and then BrC is "attributed" by difference. Yes, the actual AAE for BC is predicted to be larger than one by Mie theory when "large" particles (~100 nm, as opposed to <30 nm spherules) are used, and as has been discussed many times prior to the Liu et al. study, most notably by Lack and Langridge (Lack and Langridge, 2013) who address specifically the question of how assumptions regarding the AAE of BC influence the attribution of BrC. This is, in some ways, exactly our point in indicating that if the AAE for BC is > 1 (the base assumption, commonly made) then our attribution would overestimate the absorption attributable to BrC.

Line 402: A more fundamental reason the PALMS and SPLAT II are not capable of quantifying contributions from BC internally mixed on dust particles is the matrix effects associated with each instrument, such that the mass fractions reported on individual particles are not quantitative.

We have modified this sentence to also indicate that matrix effects may preclude explicit quantification even if particles were sampled across the entire size range. Specifically, "Although informative, these measurements unfortunately cannot be used to quantitatively assess the relative contributions of the different absorbing particle types to the supermicron absorption because both instruments sample only over a subset of the entire supermicron size range, e.g. the SPLAT-II only up to $d_{v,a} \sim 2$ μm, *and matrix effects can impact quantification of individual components in mixed particles (e.g. BC mixed with dust).*"

Line 453: Shouldn't the qualitative influence of sea salt particles be evident in the single particle mass spectrometer data?

Ideally, yes. However the particle sampling statistics during CARES make it difficult to develop a qualitative picture that can be confidently assessed in this case.

Line 514 (and elsewhere): "consistent with that of (Russell et al., 2010)" should read "consistent with that of Russell et al. (2010)"

These have been fixed.

Line 531: This opening statement should be made more descriptive, as it isn't specified what exactly the effect on climate is. It may also be worth mentioning that this is an important parameter for remote sensing retrievals.

The statement now reads:

"The extent to which particles scatter light in the backward versus forward direction has an important controlling influence on their climate impacts, *namely the amount of incident solar radiation that is reflected back to space and the associated radiative forcing* (Haywood and Shine, 1995). *Furthermore, the backscatter fraction and asymmetry parameter are important parameters for remote sensing retrievals.*"

Line 573: Was any correction for thermophoretic losses invoked? If not, the authors should comment on how much of an issue they believe this to be.

Yes, a correction for thermophoretic losses was applied, similar to what was done in Cappa et al. (2012). The correction factor for particles that passed through the TD was 0.8. We have added the following statement to the methods: "Measurements made through the thermodenuder were corrected for thermophoretic losses (Huffman et al., 2008).

Line 593: According to conventional wisdom, supermicron sulphate and nitrate tends to be in the form of salts of calcium and sodium rather than ammonium, which makes them quite involatile, so I would doubt that this is significant.

We have clarified this to indicate that we mean that there could *potentially* be evaporation of "inorganics such as *ammonium* sulfate and *ammonium* nitrate," although we agree with the reviewer that this is likely to be less important than organics.

Line 599: A related hypothesis could be that undenuded sea salt particles do not completely effloresce during drying due to the presence of magnesium salts (doi: 10.5194/acp-15-11273-2015) and organics. If the denuded particles are more completely dried out (due to the water being boiled off) then this would increase their apparent hygroscopicity further.

This is certainly a possibility. The "2-D Area Ratio" reported in the reference given (Gupta et al., 2015) for pure MgCl2 at 30% RH is about 1.3 and for pure NaCl at 80% RH is about 5. Given that the diameter growth factor (GF) for NaCl is ~2 at 80% RH, this implies that the equivalent MgCl2 GF at 30% RH is around 1.07. The mass ratio between Mg and Na in sea water is about 0.12. Thus, the MgCl2 would be only a small fraction of the sea salt (NaCl would dominate) and the MgCl2 would have only a negligible impact on the average water content of the particles. In fact, this is apparent in Gupta et al., who find that the 2-D Area Ratio is only about 1.2 at 30% RH for mixed NaCl/MgCl2 particles where the NaCl mole fraction is 0.9. In this case, the equivalent GF would be 1.05. Thus, although residual water due to magnesium salts could influence the measurements, the magnitude of this influence would be small. Although we believe this to most likely be an unimportant effect, we have added the following sentences to indicate it as a possibility. "It is possible that the thermodenuding removed residual water that did not fully evaporate in the drier due to the presence of magnesium sea salts (which do not effloresce until very low RH), leading to an apparent increase in the hygroscopicity of the thermodenuded particles. However, the potential impact from this is expected to have been quite limited, given the small amount of residual water retained by magnesium salts at low RH (Gupta et al., 2015)."

Line 620: How does this fit in with the conclusions of Doi:10.1038/Ngeo2220?

Saleh et al. (2014) also observe that the brown carbon fraction from biomass burning samples has a higher effective absorptivity at shorter wavelengths. Saleh et al. (2014) also argue that heating can leave behind more lower-volatility organics that are more absorbing than the higher-volatility species that evaporate more easily in a thermodenuder. This can lead to a suppression of observable absorption enhancement values over the "true" value for BC. However, Cappa et al. (2011) showed that the thermodenuder-derived $E_{abs}$ values behaved very similarly to $E_{abs}$ values derived from consideration of mass absorption coefficients for CARES, which suggests that such "residual" more-absorbing organics are playing a negligible role here (see the Supplementary Material of that paper). We also note that Zhang et al. (2016) also found good agreement between thermodenuder-derived and MAC-derived $E_{abs}$ values for a region known to be impacted by biomass burning (the focus of the Saleh et al. study), suggesting that the limitations of the heating method identified by Saleh et al. (2014) may be limited in scope. This is certainly an issue that remains to be fully resolved, but is not a major point of this work and thus we have not pursued it further here.

Line 646: The caveat should be added that this is assuming that the particles have not acquired an involatile coating (e.g. coagulation with sea salt, condensation of humiclike SOA), because this could confound any attempt to isolate the effect of morphological changes on SSA.

We have added the following sentence: "However, if non-absorbing and non-volatile materials remained (e.g. sea salt), then the extrapolated SSA would not be fully representative of pure BC particles, confounding straightforward interpretation in terms of morphological changes."

Line 653: It should be noted that if the particles are thought to be very non-spherical, the SMPS sizing is likely to be overestimated to a large degree

True, but shape effects would not be large enough to make the SMPS-measured 300 nm shift down to < ~50 nm, which would be necessary to explain such small SSA values.

Line 653: If the particles are fundamentally different to 'normal' black carbon particles, then the SP2's calibration could be invalid and the equivalent core sizes reported inaccurate. If the instrument had a narrow band incandescence detector, it may be informative to compare the ratio of this to the broadband detector to see if the apparent colour temperature had changed.

This is an interesting point. Laborde et al. (2012) reported SP2 sensitivities for a variety of different BC types. The largest difference observed corresponded to a sensitivity ratio of about 1.6. If the SP2 were too sensitive towards the asphalt particles by a factor of 1.6 (in per particle mass), then the diameter would be overestimated by a factor of $1.6^{1/3} = 1.17$. Fig. S9 (now S10) showed the particles during the asphalt-impacted period to be about a factor of 2 larger in diameter. This would require a mass sensitivity difference of $2^3 = 8$. This seems unreasonably large, given the results shown in Laborde et al. (2012). Thus, although we agree that the particles may have been detected with a somewhat different sensitivity than typical ambient BC particles, the difference is unlikely to make a substantial impact on the conclusions here, and certainly unlikely to invalidate our conclusion that it is most likely that the particles are composed of smaller spherules.

Figure S1: Adjust the colour scale so that areas covered by land are green rather than blue.

Done.

Figure S9: Given that Babs of black carbon is more closely related to mass than number, a mass-weighted distribution comparison would be informative here.

Done. The conclusions are unaffected.

To be clear, Kassianov et al. (2012) calculated supermicron aerosol extinction from size distributions, while in the current study we actually report on measurements of the scattering, absorption and extinction by the supermicron particles in this region. As we now note in the introduction, direct measurements of supermicron optical properties are fairly uncommon, and where they do exist are most often found for the marine boundary layer where the contribution of sea spray will be large. We have focused our revisions on studies in which optical properties were measured (either *in situ* or remote sensing) and not considered measurements of particulate mass concentrations alone. To this end, we have added an extensive comparison with some remote sensing results that look at the relationship between Angstrom exponents and the fine mode fraction of extinction, and that look at the relationship between the SSA and the fine mode fraction of extinction. We have added two new figures in support, one in the supplemental and one in the main text.

The added text and figures are provided below.

[revised manuscript text omitted]

*Minor comments:*

Line 129: Give model and manufacturer for SMPS.

Done.

Line 176: What is the part number of the NOx chemiluminescence instrument?

We now give further information about the operation of the instrument, specifically stating "Gas-phase concentrations of the sum of NO and $NO_2$ (= $NO_x$) and the sum of nitrogen oxides (= $NO_y$) were measured using a 2-channel chemiluminescence instrument (Air Quality Design, Inc, High Performance, 2-Channel) in which $NO_2$ is photolyzed to NO *using a blue light photolytic converter* and $NO_y$ is converted to NO on a Mo catalyst," and where the new text is in italics.

*Typographic corrections:*

Line 32: "... but the there is some" - done

Line 79: Use SI units. - done

Lines 99,125, 203, 205: Remove comma after June - done

Line 157: This is a very long week. – We now indicate this is a week and a half

**References:**

[revised manuscript text omitted]

The Supplemental Material contains three tables (Table S1-S3) and eleven figures (Figure S1-S11)

to provide additional information about and support for the results presented in the manuscript main text.

**Table S1.** Summary of abbreviations and symbols used in this work.

| Symbol | Definition |
|---|---|
| $PCA_{NOx}$ | Average photochemical age of the air mass |
| $PCA_{HC}$ | Average photochemical age of the air mass |
| $PM_1$ | Particulate matter with aerodynamic diameter < 1 μm |
| $PM_{10}$ | Particulate matter with aerodynamic diameter < 10 μm |
| $PM_{super}$ | Particulate matter with aerodynamic diameter between 1 and 10 μm |
| $b_{sca,super}$ | Scattering attributed to particles between 1 and 10 μm |
| $b_{abs,super}$ | Absorption attributed to particles between 1 and 10 μm |
| $f_{sca,PM1}$ | Fraction of scattering from particles < 1 μm diameter (submicron) |
| $f_{abs,PM1}$ | Fraction of absorbing from particles < 1 μm diameter (submicron) |
| $f_{sca,super}$ | Fraction of scattering from particles between 1 and 10 μm diameter |
| $f_{abs,super}$ | Fraction of absorbing from particles between 1 and 10 μm diameter |
| $AAE_{\lambda1,\lambda2}$ | Absorbing Ångstrom exponent from λ1 and λ2 |
| $AAE_{PM1}$ | Absorption Ångstrom exponent for particles < 1 μm diameter (submicron) |
| $AAE_{PM10}$ | Absorption Ångstrom exponent for particles < 10 μm diameter |
| $\Delta AAE_{10-1}$ | Absorption Ångstrom exponent for particles from 1 to 10 μm diameter |
| $\Delta AAE_{amb-TD}$ | Difference in AAE between ambient and thermodenuded states |
| $SAE_{\lambda1,\lambda2}$ | Scattering Ångstrom exponent from a given wavelength pair (λ1 and λ2) |
| $SAE_{PM1}$ | Scattering Ångstrom exponent for particles < 1 μm diameter (submicron) |
| $SAE_{PM10}$ | Scattering Ångstrom exponent for particles < 10 μm diameter |
| $\Delta SAE_{10-1}$ | Difference in scattering Ångstrom exponent for particles from 1 to 10 μm diameter |
| $EAE_{PM10}$ | Extinction Ångstrom exponent for particles < 10 μm diamete |
| $EAE_{PM10}$-50 | $EAE_{PM10}$ value where $f_{sca,PM1}$ = 0.5 |
| $SSA$ | Single scatter albedo – fraction of extinction due to scattering |
| $f_{bsca}$ | Fraction of light that is backscattered |
| $g_{sca}$ | Asymmetry parameter |
| $f_{ext,TD}$ | Fraction of extinction remaining in the thermodenuder |
| $\gamma_{RH}$ | Hygroscopicity |
| $\Delta\gamma_{RH}$ | Difference in hygroscopicity between undenuded and denuded |
| $MAC_{PM1}$ | Mass absorption coefficient for particles < 1 μm diameter |
| $MAC_{PM10}$ | Mass absorption coefficient for particles < 10 μm diameter |
| $MAC_{super}$ | Mass absorption coefficient for particles from 1 to 10 μm diameter |
| $MSC_{PM1}$ | Mass scattering coefficient for particles < 1 μm diameter |
| $MSC_{PM10}$ | Mass scattering coefficient for particles < 10 μm diameter |
| $MSC_{super}$ | Mass scattering coefficient for particles from 1 to 10 μm diameter |
| $E_{abs}$ | Absorption enhancement |
| $AOD$ | Aerosol Optical Depth |
| $FMF$ | Fine mode fraction of extinction |

**Table S2.** Results from linear fitting of $SAE_{PM10}$ versus $f_{sca,PM1}$.

| Wavelength Pair $SAE_{PM10}$ | Wavelength $f_{sca,PM1}$ (nm) | Slope[*] | Intercept[#] |
|---|---|---|---|
| 450-550 | 450 | 2.65 | -0.34 |
| 450-550 | 550 | 2.69 | -0.05 |
| 450-550 | 700 | 2.81 | 0.27 |
| 450-700 | 450 | 2.63 | -0.41 |
| 450-700 | 550 | 2.70 | -0.14 |
| 450-700 | 700 | 2.92 | 0.15 |
| 550-700 | 450 | 2.62 | -0.47 |
| 550-700 | 550 | 2.71 | -0.21 |
| 550-700 | 700 | 3.01 | 0.05 |
|  | T1 |  |  |
| 450-550 | 450 | 2.62 | -0.24 |
| 450-550 | 550 | 2.66 | -0.06 |
| 450-550 | 700 | 2.76 | 0.40 |
| 450-700 | 450 | 2.69 | -0.42 |
| 450-700 | 550 | 2.76 | -0.14 |
| 450-700 | 700 | 2.97 | 0.18 |
| 550-700 | 450 | 2.77 | -0.59 |
| 550-700 | 550 | 2.85 | -0.31 |
| 550-700 | 700 | 3.15 | -0.02 |
| 550-700 | 700 | 3.01 | 0.05 |

[*] The fit uncertainties were typically < 0.02, which is much smaller than the experimental uncertainty.
[#] The fit uncertainties were typically < 0.01, which is much smaller than the experimental uncertainty.

**Table S3.** Summary of the $EAE_{PM10}$-50 values dependence on the wavelength pairs used. See Figure S6

for the relevant observations.

| $EAE_{PM10}$ Wavelength Pair (nm) | $f_{ext,PM1}$ Wavelength (nm) | $EAE_{PM10}$-50 |
|---|---|---|
| 450,550 | 450 | 0.92 |
| 450,550 | 550 | 1.15 |
| 450,550 | 700 | 1.42 |
| 450,700 | 450 | 0.85 |
| 450,700 | 550 | 1.09 |
| 450,700 | 700 | 1.37 |
| 550,700 | 450 | 0.79 |
| 550,700 | 550 | 1.03 |
| 550,700 | 700 | 1.32 |

[Figure]

**Figure S1.** (left) A map of California, showing the general measurement location. (right) A closer-up view of the two observational sites, T0 near Sacramento, CA and T1 near Cool, CA. The gray lines show the main interstate and highway network. Dark blue areas indicate water.

[Figure]

**Figure S2.** Schematic of the sampling scheme during CARES for the (left) T0 site in Sacramento and (right) T1 site in Cool.

[Figure]

**Figure S3.** Merged campaign-average mobility-equivalent size distribution for the T0 site showing the measurements made
using the SMPS (red) and APS (black). The APS aerodynamic diameters were adjusted to mobility-equivalent diameters
assuming a material density of 2.0 g cm$^{-3}$. The number-weighted distribution is shown as solid lines and the volume-weighted
size distribution as dashed lines.

[Figure]

**Figure S4.** The relationship between the scattering Ångstrom exponent for PM$_{10}$ for different wavelength pairs and the PM$_1$/PM$_{10}$ scattering ratio ($f_{\text{sca,PM1}}$) at different wavelengths for the T0 site. The points are colored according to time during the campaign. Slope and intercept values from the linear fits (black lines) are reported in Table S2.

[Figure]

**Figure S5.** The relationship between the scattering Ångstrom exponent for PM₁₀ for different wavelength pairs and the PM₁/PM₁₀ scattering ratio ($f_{sca,PM1}$) at different wavelengths for the T1 site. The points are colored according to time during the campaign. Slope and intercept values from the linear fits (black lines) are reported in Table S2.

| Deleted: 4 |
| Deleted: 1 |

[Figure]

Figure S6. The relationship between the extinction Ångstrom exponent for $PM_{10}$ for different wavelength pairs and the $PM_1/PM_{10}$ extinction ratio ($f_{ext,PM1}$) at different wavelengths for the T0 site. The points are colored according to time during the campaign, as in Figure S4. The $EAE$ values when $f_{ext,PM1} = 0.5$ are reported in Table S3.

[Figure]

**Figure S7.** Fractional number abundance of supermicron particles ($d_{va} > 1$ micron) as measured by the SPLAT-II instrument
at the T0 site. It should be noted that the upper-limit sampling range for SPLAT-II is around 2 microns, and thus supermicron
particles that are larger than this are not characterized.

[Figure]

**Figure S8.** Calculated relationship between the mass scattering coefficient and mobility diameter for spherical particles with
density = 2 g cm$^{-3}$. A complex refractive index of RI = 1.5 + 0.0$i$ was used.

[Figure]

**Figure S9.** The central panels show time-series of the mass scattering coefficient and PM concentration for supermicron particles for both T0 (black lines) and T1 (colored lines). The outer panels show back trajectories calculated form HYSPLIT, using the NAM meteorological data, arriving at the T0 site at noon local time each day. Each outer panel shows three back trajectories that are separated by 24 hours (note that time goes backwards in these panels). The boxes around each outer panel correspond in color to the boxes shown in each of the central panels and provide a visual reference as to which trajectories correspond to which time period.

[Figure]

**Figure S10.** The relationship between the supermicron particle concentration (a,b) or the absolute supermicron absorption at 532 nm (c,d) and the surface-weighted median diameter of the supermicron mode. Observations from T0 are shown in (a) and (c) and from T1 in (b) and (d). For reference, the time series of [PM$_{super}$] is shown for both sites in panel (e), where T0 is shown in black and T1 in color. The points are colored according to date, as indicated in the color scale.

[Figure]

[Figure]

**Figure S11.** Refractory BC (a) number-weighted and (b) mass-weighted size distributions measured by the SP2 during the asphalt impacted period (22 June between 1:15 am and 2:30 am, local time) and during typical time periods. The typical size distribution is shown as a dashed blue line and the asphalt-impacted period as a red line.